# Charge self-regulation in 1T'''-MoS$_2$ structure with rich S vacancies for enhanced hydrogen evolution activity

Xiaowei Guo[1,2,7], Erhong Song [1,2,7], Wei Zhao[1,2,7], Shumao Xu[1], Wenli Zhao[3], Yongjiu Lei[4], Yuqiang Fang[1,2], Jianjun Liu [1,2,5] ✉ & Fuqiang Huang [1,2,6] ✉

Active electronic states in transition metal dichalcogenides are able to prompt hydrogen evolution by improving hydrogen absorption. However, the development of thermodynamically stable hexagonal 2H-MoS$_2$ as hydrogen evolution catalyst is likely to be shadowed by its limited active electronic state. Herein, the charge self-regulation effect mediated by tuning Mo−Mo bonds and S vacancies is revealed in metastable trigonal MoS$_2$ (1T'''-MoS$_2$) structure, which is favarable for the generation of active electronic states to boost the hydrogen evolution reaction activity. The optimal 1T'''-MoS$_2$ sample exhibits a low overpotential of 158 mV at 10 mA cm$^{-2}$ and a Tafel slope of 74.5 mV dec$^{-1}$ in acidic conditions, which are far exceeding the 2H-MoS$_2$ counterpart (369 mV and 137 mV dec$^{-1}$). Theoretical modeling indicates that the boosted performance is attributed to the formation of massive active electronic states induced by the charge self-regulation effect of Mo−Mo bonds in defective 1T'''-MoS$_2$ with rich S vacancies.

Molybdenum disulfide with hexagonal structure (2H-MoS$_2$) has been considered as a promising substitution to precious-metal catalyst for hydrogen evolution reaction (HER) due to its low cost and moderate activity[1–3]. Theoretical calculations and experimental observations reveal that the HER activity of 2H-MoS$_2$ is mainly ascribed to the active electronic states around Fermi level that optimizes the strength of hydrogen adsorption[4–6]. However, the conventional 2H-MoS$_2$ is short of enough active electronic states due to the inert basal planes and limited Mo/S-terminated edge sites, limiting the overall catalytic performance. Up to date, several activation and optimization strategies have been conducted to regulate the electronic states such as heterogeneous atom doping[7–17], defect engineering[18–27], phase transformation[28–34] and domain boundaries[35–37]. However, the electrocatalytic performance of MoS$_2$ is still unsatisfactory and inferior to that of noble metals like Pt,

etc. It is of much importance to design more efficient strategies for maximizing active sites and regulating active electronic states.

The bulk 2H-MoS$_2$ is intrinsically an indirect-gap semiconductor (band gap $E_g$ = 1.29 eV; electronic conductivity at room temperature $\sigma_{RT}$ = ~10$^{-4}$ S cm$^{-1}$) and has insufficient active electronic states. In contrast, a discovered metastable trigonal 1T'''-MoS$_2$ is a narrow-gap semiconductor ($E_g$ = 0.65 eV and $\sigma_{RT}$ = 2.26 S cm$^{-1}$) with the corner-sharing Mo$_3$ triangular trimers[30,38]. The rich Mo−Mo bonds in the 1T'''-MoS$_2$ structure may serve as electron reservoirs to regulate the electronic states across the Fermi surface to control hydrogen adsorption/desorption. 1T'''-MoS$_2$ is expected to be a promising HER electrocatalyst.

Herein, we propose a concept of charge self-regulation to manipulate the active electronic states across the Fermi surface of 1T'''-MoS$_2$ with S vacancies (1T'''-MoS$_2$-V$_S$) to boost the HER activity. To

[1]State Key Laboratory of High Performance Ceramics and Superfine Microstructure, Shanghai Institute of Ceramics, Chinese Academy of Sciences, 200050 Shanghai, China. [2]Center of Materials Science and Optoelectronics Engineering, University of Chinese Academy of Science, 100049 Beijing, China. [3]School of Physical and Mathematical Sciences, Nanjing Tech University, 211800 Nanjing, China. [4]Materials Science and Engineering, King Abdullah University of Science and Technology (KAUST), Thuwal 23955-6900, Saudi Arabia. [5]Shanghai Institute of Materials Genome, 99 Shangda road, 200444 Shanghai, China. [6]State Key Laboratory of Rare Earth Materials Chemistry and Applications, College of Chemistry and Molecular Engineering, Peking University, 100871 Beijing, China. [7]These authors contributed equally: Xiaowei Guo, Erhong Song, Wei Zhao. ✉e-mail: jliu@mail.sic.ac.cn; huangfq@mail.sic.ac.cn

verify the charge self-regulation effect, 1T'''-MoS$_2$-V$_S$ bulk crystals were synthesized by topochemical reaction and chemical etching in acidic K$_2$Cr$_2$O$_7$ aqueous solution. S vacancies are precisely introduced to activate the Mo−Mo bonds through the formation of dangling bonds. The activated Mo−Mo bonds can self-regulate the electronic states to promote proton adsorption by enhancing the interaction between S and H atoms (S−H bonds). The optimal 1T'''-MoS$_2$-V$_S$ exhibits remarkable HER performance with an overpotential of 158 mV at 10 mA cm$^{-2}$, a Tafel slope of 74.5 mV dec$^{-1}$ and excellent cycling stability, far exceeding 2H-MoS$_2$ with S vacancies (2H-MoS$_2$-V$_S$: 369 mV and 137 mV dec$^{-1}$). This research highlights that the inherent metal-metal bonds in TMDs enable the regulation of the active electronic states to perform highly efficient electrocatalytic hydrogen evolution.

## Results

### Effect of electronic states toward HER activity

The basal plane activation of MoS$_2$ is clarified in Fig. 1a–c, illustrating the difference in active electronic states of 2H-MoS$_2$, 2H-MoS$_2$-V$_S$ and 1T'''-MoS$_2$-V$_S$, respectively. As the Fermi surface is isolated, the absence of active electronic state results in the inertia of the planes of 2H-MoS$_2$ (Fig. 1a). The activated $p$ orbital of S mainly hybridizes with Mo $d$ orbital of 2H-MoS$_2$-V$_S$ at relatively deep energy level, which is not beneficial to provide extra electronic states around the Fermi surface for proton adsorption (Fig. 1b). In contrast, the active electronic states of Mo−Mo bonds and S atoms across the Fermi surface in 1T'''-MoS$_2$-V$_S$ lead to the charge transfer from activated Mo−Mo bonds to S atoms (Fig. 1c). The active Mo−Mo bonds induce the charge redistribution and tune the electronic state of S atoms, which can enhance the strength of S−H bond. Similar to our previously reported results[39], the unique 1T'''-MoS$_2$-V$_S$ with intrinsically active Mo−Mo bonds has spontaneous optimization effect on active electronic states. S atoms near Mo−Mo bonds in the basal plane of 1T'''-MoS$_2$-V$_S$ become electrocatalytically HER active sites.

To investigate the charge self-regulation effect on activating Mo−Mo bonds to improve HER activity, different concentrations of S

vacancies in the 1T'''-MoS$_2$ were controlled by using chemical etching method. The schematic synthesis of 1T'''-MoS$_2$-V$_S$ from the KMoS$_2$ and crystals is shown in Fig. 2a. Highly ordered K$^+$ ions are occupied in the interlayer of KMoS$_2$ crystal, which is synthesized by high temperature solid-state reaction. Mo−Mo bonds in KMoS$_2$ partially remain in bulk 1T'-MoS$_2$ and 1T'''-MoS$_2$ upon the extraction of K$^+$ ions by oxidizing Mo (III) to Mo (IV)[40]. The 1T'''-MoS$_2$ was further oxidized in acidic K$_2$Cr$_2$O$_7$ solution to obtain different concentrations of S vacancies (1T'''-MoS$_2$-V$_S$: V$_S$ = 2.0%, 7.7%, 10.6%, 17.9%, 22.9%) via different etching times (0 h, 0.5 h, 1 h, 2 h, 3 h). The 2H-MoS$_2$-V$_S$ was prepared by high-temperature vacuum annealing of 1T'''-MoS$_2$−10.6%. Therefore, the adjustable S vacancies via a simple chemical etching method in this work is expected to explore the charge self-regulation effect of Mo−Mo bonds on HER activity.

### Phase and structural characterization

Crystal structure data of pristine 1T'''-MoS$_2$ were measured by the single crystal X-ray diffraction (XRD). The top and side views of the atomic structure of 1T'''-MoS$_2$ have been shown in the Supplementary Fig. 1. As shown in the scanning electron microscopy (SEM) images (Supplementary Fig. 2), etching time has little effect on the morphology and size of 1T'''-MoS$_2$-V$_S$, and the lateral size of 1T'''-MoS$_2$-V$_S$ sheets is about 100 μm. The energy dispersive X-ray spectroscopy results (Supplementary Fig. 3) indicate that no residual K element was observed in 1T'''-MoS$_2$−10.6%.

X-ray diffraction (XRD) patterns (Supplementary Fig. 4) demonstrated the successful preparation of 1T'-MoS$_2$ and 1T'''-MoS$_2$, which is further confirmed by the selected area electron diffraction patterns (Supplementary Fig. 5). The prepared 1T'''-MoS$_2$ crystallizes in the trigonal space group P31m with the lattice parameters $a$ = 5.580 Å and $c$ = 5.957 Å[41], as listed in the Supplementary Tables 1–3. The Mo−Mo bonds (3.013 Å) of 1T'''-MoS$_2$ form corner-sharing Mo$_3$ trimers in the $ab$ plane. The distortion of [MoS$_6$] octahedral coordination and the varied Mo−S bond lengths (2.370, 2.521, 2.459, and 2.298 Å) in 1T'''-MoS$_2$ result in three types of S sites. The magnified XRD patterns (13–16°) of the different samples (Supplementary Fig. 6) show that the highest peak shifts to a lower angle with the increase of S vacancies, maybe due to interlayer spacing expansion between S – Mo-S layers[42].

Figure 2b shows the Raman spectra of 1T'''-MoS$_2$-2.0%, 1T'''-MoS$_2$−10.6% and the reference 2H-MoS$_2$-V$_S$ (obtained by thermal treatment of 1T'''-MoS$_2$−10.6%), respectively. From the detailed vibrational modes of the Raman spectra (Fig. 2b), six characteristic peaks of 1T'''-MoS$_2$-2.0% are located at 177 ($J_1$), 243, 267 ($J_3$), 305 ($E_{1g}$), 398 ($A_{1g}$) and 463 cm$^{-1}$, consistent with previously reported results[41,43]. With the increase of S vacancies, the $E_{1g}$ peak of 1T'''-MoS$_2$−10.6% shifts to 302 cm$^{-1}$. The Raman spectrum of 1T'-MoS$_2$ shows four characteristic peaks located at 150 ($J_1$), 212($J_2$), 280 ($E_{1g}$) and 324 ($J_3$) cm$^{-1}$ (Supplementary Fig. 7)[30,44–46]. The $E_{1g}$ peak (383 cm$^{-1}$) of reference 2H-MoS$_2$-V$_S$ is completely different from the characteristic peaks in the 1T'''-MoS$_2$-V$_S$ samples due to their different crystal structures[47].

### Chemical state and defect concentration

The local bond lengths and coordination environment of 1T'''-MoS$_2$-V$_S$ (V$_S$ = 2.0%, 10.6%) were studied by X-ray absorption spectroscopy (XAS) with Mo foil and commercial 2H-MoS$_2$ as reference. The Mo K-edge X-ray adsorption near edge structure (XANES) spectra (Fig. 2c) indicate that 1T'''-MoS$_2$-2.0% and 1T'''-MoS$_2$−10.6% possess similar absorption edge and white line peaks, but different from 2H-MoS$_2$ and Mo foil. The oxidation state of Mo element in Mo foil is zero while in 2H-MoS$_2$ is +4. Figure 2d shows the result of Fourier transform of the extended X-ray absorption fine structure (FT-EXAFS) oscillation. The two characteristic peaks at 1.9 and 2.8 Å (not corrected by scattering phase shift) can be assigned to the nearest Mo−S and Mo−Mo bonds, respectively[48]. No obvious shift is observed in 1T'''-MoS$_2$−10.6% with higher S vacancy concentration, indicating Mo−Mo bond length is

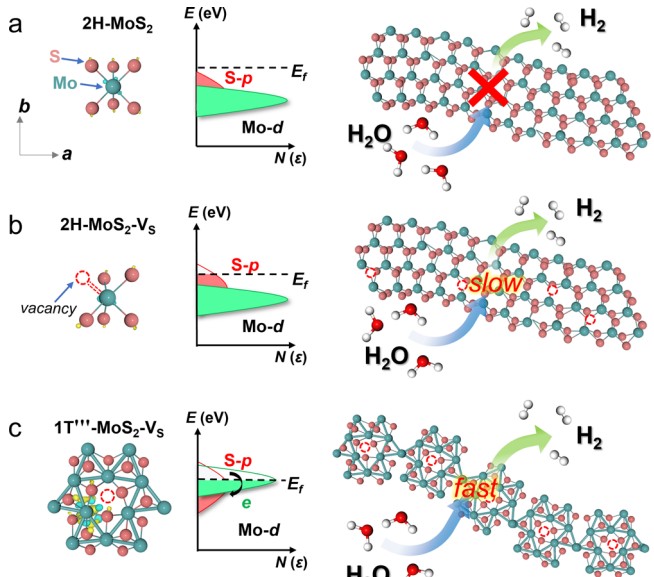

**Fig. 1 | Schematic illustration of charge self-regulation effect on manipulating active electronic states. a** 2H-MoS$_2$, (**b**) 2H-MoS$_2$-V$_S$ without Mo−Mo bonds (Only gap states around the Fermi level induced by S atom) and (**c**) 1T'''-MoS$_2$-V$_S$ with Mo−Mo bonds (Charge transfers from activated Mo−Mo bonds to S atoms due to uplifted states of Mo−Mo bonds). H$_2$O can be electrocatalytically reduced to H$_2$ through 2H-MoS$_2$-V$_S$ and 1T'''-MoS$_2$-V$_S$ under ambient conditions. The yellow and blue regions indicate the accumulation of positive and negative charge, respectively.

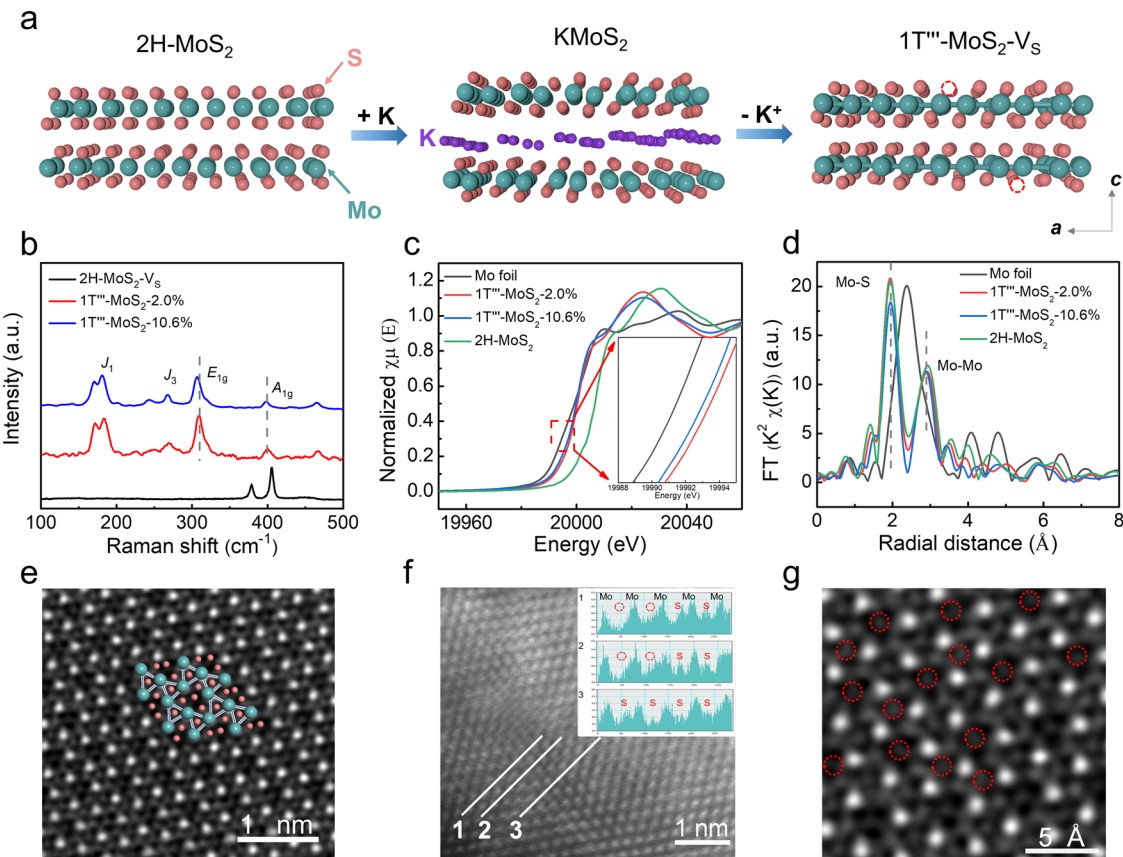

**Fig. 2 | Characterizations of structure of 1T'''-MoS$_2$-V$_S$. a** Fabrication procedure of 1T'''-MoS$_2$-V$_S$ crystals synthesized by high temperature solid-state reactions for chemical insertion and wet chemical routes for K$^+$ extraction and S vacancies generation. **b** Raman spectra of 2H-MoS$_2$-V$_S$, 1T'''-MoS$_2$–2.0% and 1T'''-MoS$_2$–10.6%. **c** Mo K-edge XANES spectra and (**d**) FT-EXAFS spectra of Mo foil, 2H-MoS$_2$, 1T'''-MoS$_2$–2.0% and 1T'''-MoS$_2$–10.6%. **e** HAADF-STEM image of 1T'''-MoS$_2$–10.6% sheet.

**f** ACTEM observations of S vacancies in a 1T'''-MoS$_2$–10.6% sheet. The corresponding intensity distribution follows three different lines. Molybdenum atom has higher contrast than sulfur atom because of its high atomic number. The red dashed circles indicate a decrease in intensity due to the absence of S atoms. **g** HAADF-STEM image of 1T'''-MoS$_2$ crystal with -10.6% S-vacancy. The biggest and brightest dots are Mo atoms. Red dotted circles correspond to S-vacancies.

nearly unchanged before and after chemical etching. The characteristic peak (2.9 Å) of 2H-MoS$_2$ can be attributed to longer Mo−Mo distance. However, the reduced peak intensity of Mo−S bonds and unchanged peak intensity of Mo−Mo bonds in 1T'''-MoS$_2$–10.6% manifests the decrease of coordination number, probably due to the generation of more defects after increasing S vacancies. More S defects in 1T'''-MoS$_2$–10.6% result in a decrease in Mo K-edge oscillation functions k$^3_\chi$ (k) at the k range of 0–14 Å$^{-1}$ (Supplementary Fig. 8) compared with 1T'''-MoS$_2$–2.0% and 2H-MoS$_2$. Furthermore, the Mo L$_3$-edge XANES spectra of 2H-MoS$_2$ and 1T'''-MoS$_2$–10.6% were compared (Supplementary Fig. 9). For 2H-MoS$_2$, the Mo L$_3$-edge white line peak locates at 2524.01 eV and it is ascribed to the Mo$^{4+}$. The characteristic peak shifts 0.45 eV to lower energy, indicating lower valence state Mo in 1T'''-MoS$_2$–10.6%[49,50]. The result is consistent with Fig. 2c.

In Fig. 2e, the lattice structure of 1T'''-MoS$_2$–10.6%, a highly distorted structure with a $\sqrt{3}$a × $\sqrt{3}$a superlattice, is identified by the high-angle annular dark-field scanning transmission electron microscopy (HAADF-STEM). The presence of S vacancies in the 1T'''-MoS$_2$–10.6% is examined by aberration-corrected transmission electron microscopy (ACTEM) shown in Fig. 2f. Lines 1 and 2 show two S gaps, while the area marked in line 3 shows no defects, indicating the presence of S vacancies in the 1T'''-MoS$_2$–10.6%. The HAADF-STEM image of 1T'''-MoS$_2$–10.6% (Fig. 2g) further confirms the successful formation of S-vacancies with corresponding concentrations.

Room-temperature electron paramagnetic resonance (EPR) spectroscopies for 1T'''-MoS$_2$-V$_S$ were carried out to measure the concentrations of S vacancies. The characteristic EPR peaks of S vacancies

(Supplementary Fig. 10) locate at $g = 2.003$ in all samples, proving the existence of S vacancies. The S signal can be attributed to the Mo−S dangling bond and peak strength of the S signal is proportional to the concentration of S vacancies[51]. Based on the increase in signal intensity with the prolonged etching time, EPR results confirm that concentrations of S vacancies are positively correlated with oxidation time. As shown in Supplementary Figs. 11 and 12, XPS was used to study the change in the chemical state of elements. Two peaks of 1T'''-MoS$_2$–10.6% are located at 228.5 and 231.7 eV, corresponding to the Mo$^{4+}$ 3d$_{5/2}$ and 3d$_{3/2}$. The Mo 3d peaks of reference 2H-MoS$_2$-V$_S$ are absent in 1T'''-MoS$_2$–10.6%, confirming the purity of 1T''' phase. The S 2p spectrum of 1T'''-MoS$_2$–10.6% shows two peaks at 163.3 and 162.15 eV, which belong to S 2p$_{1/2}$ and S 2p$_{3/2}$, respectively. No peaks assigned to the oxidized Mo and S (hexavalent Mo, elemental S or high-valence S component) species appear. Compared to the reference 2H-MoS$_2$-V$_S$, S 2p orbital peaks of 1T'''-MoS$_2$–10.6% shift 0.62 eV toward lower binding energies, suggesting the partial charges from the activated Mo−Mo bonds are transferred to neighboring S atoms. By measuring the signals of Mo 3d and S 2p regions, the ratio of S and Mo (Supplementary Table 4) of each region is converted to the concentration of S vacancies.

The electrical properties of 1T'''-MoS$_2$–10.6% and 2H-MoS$_2$-V$_S$ are tested by the physical property measurement system (Supplementary Fig. 13). The electrical conductivity at 298 K of 2H-MoS$_2$-V$_S$ and 1T'''-MoS$_2$–10.6% are determined to be ~10$^{-4}$ and 0.53 S m$^{-1}$, respectively. The electrical conductivity of 1T'''-MoS$_2$–10.6% is about three orders of magnitude higher than 2H-MoS$_2$-V$_S$. High electrical conductivity is beneficial for the charge transfer kinetics in HER process.

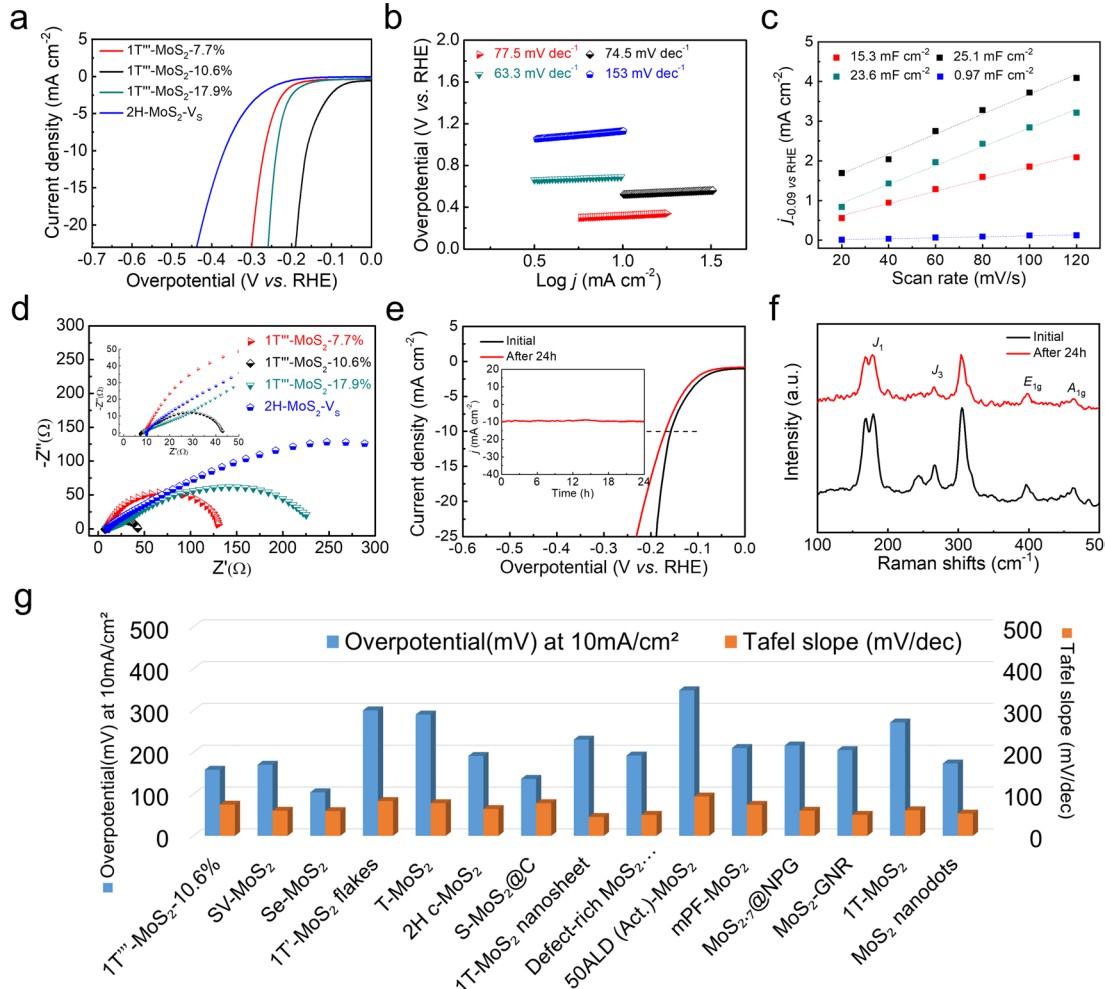

**Fig. 3 | The electrocatalytic HER properties of 1T'''-MoS₂-Vₛ.** HER performance in 0.5 M H₂SO₄. **a** Linear sweep polarization curves of 1T'''-MoS₂-Vₛ (Vₛ = 7.7%, 10.6%, 17.9%) and 2H-MoS₂-Vₛ. **b** The corresponding Tafel curves from the polarization curves. **c** Plots of current density difference against scan rates, $j_{-0.09}$ is the difference between anodic and cathodic current densities at −0.09 V (vs. RHE). In the plot, the capacitance was normalized by the geometric surface area of electrodes. **d** Nyquist plots of 1T'''-MoS₂-Vₛ (Vₛ = 7.7%, 10.6%, 17.9%) and 2H-MoS₂-Vₛ. **e** LSV curves of 1T'''-MoS₂−10.6% before and after 24 h of durability test. (Inset: the long-term durability tests at η = 158 mV) (**f**) Raman spectra of 1T'''-MoS₂−10.6% before and after 24 h test. **g** Overpotentials at current density of 10 mA cm⁻² of 1T'''-MoS₂−10.6% and 2H-MoS₂-Vₛ catalysts in comparison to values reported previously for HER catalysts in acidic electrolytes.

## Evaluation toward electrochemical hydrogen evolution

To evaluate the influence of S vacancies on HER performance, the electrocatalytic activity of 1T'''-MoS₂-Vₛ was studied in Ar-saturated 0.5 mol L⁻¹ H₂SO₄ solution. Linear sweep voltammetry (LSV) curves are shown in Fig. 3a. When the oxidation time increases up to 1 h, the overpotential is continually decreased to 158 mV (1T'''-MoS₂-10.6%), which agrees with $\Delta G_{H^*}$ of the theoretical calculations (Supplementary Fig. 14). To reveal the kinetic metrics, Tafel slope is used to investigate the rate-determining step for HER. As presented in Fig. 3b and Supplementary Fig. 15, the Tafel slope of 1T'''-MoS₂−10.6% is 74.5 mV dec⁻¹, smaller than that of 2H-MoS₂-Vₛ. This suggests the pivotal role of activated S atoms for the absorption of hydrogen. Compared with 2H-MoS₂-Vₛ, 1T'''-MoS₂-Vₛ have higher HER activity due to the self-regulation effect of Mo−Mo bonds. After activation by S vacancies, the Mo−Mo bonds enhance the S−H bonds by changing the electronic states of S atoms around Mo−Mo bonds, further leading the optimal $\Delta G_{H^*}$ closer to 0. Accordingly, both experimental and theoretical results (Supplementary Fig. 14) follow the similar trend that the 1T'''-MoS₂−10.6% has the optimal performance. Hence, the combination of proper S vacancies and Mo−Mo bonds leads to higher HER activity of 1T'''-MoS₂.

Electrochemical active surface areas (ECSAs) are estimated from the double-layer capacitance ($C_{dl}$) by measuring cyclic voltammetry (CV) curves. According to Fig. 3c and Supplementary Fig. 16, 1T'''-MoS₂−10.6% possesses the highest value of ECSA (25.1 mF cm⁻²), considerably larger than those of 1T'''-MoS₂−2.0% (15.2 mF cm⁻²), 1T'''-MoS₂−7.7% (15.3 mF cm⁻²), 1T'''-MoS₂−17.9% (23.6 mF cm⁻²), 1T'''-MoS₂−22.9% (22.9 mF cm⁻²), commercial 2H-MoS₂ (1.17 mF cm⁻²) and 2H-MoS₂-Vₛ (0.97 mF cm⁻²), suggesting proper concentration of S vacancies can significantly expand the ECSAs and expose more electrochemically active sites. Besides that, the electrode kinetics was investigated at the hydrogen evolution voltage by electrochemical impedance spectroscopy. The result corresponds to charge transfer resistance of proton between electrode and electrolyte (Fig. 3d and Supplementary Fig. 15). 1T'''-MoS₂−10.6% displays the lowest $R_{ct}$ (35.27 Ω) among all the samples, indicating appropriate S vacancies can accelerate electrode kinetics for HER and reduce ohmic loss. S vacancies in 1T'''-MoS₂ not only modify the electrical conductivity to facilitate electron transport but also tune the proton adsorption/desorption (in terms of $\Delta G_{H^*}$) to optimize HER activity.

The long-term cycling stability of various 1T'''-MoS₂-Vₛ samples was examined by polarization curves before and after 24 h of

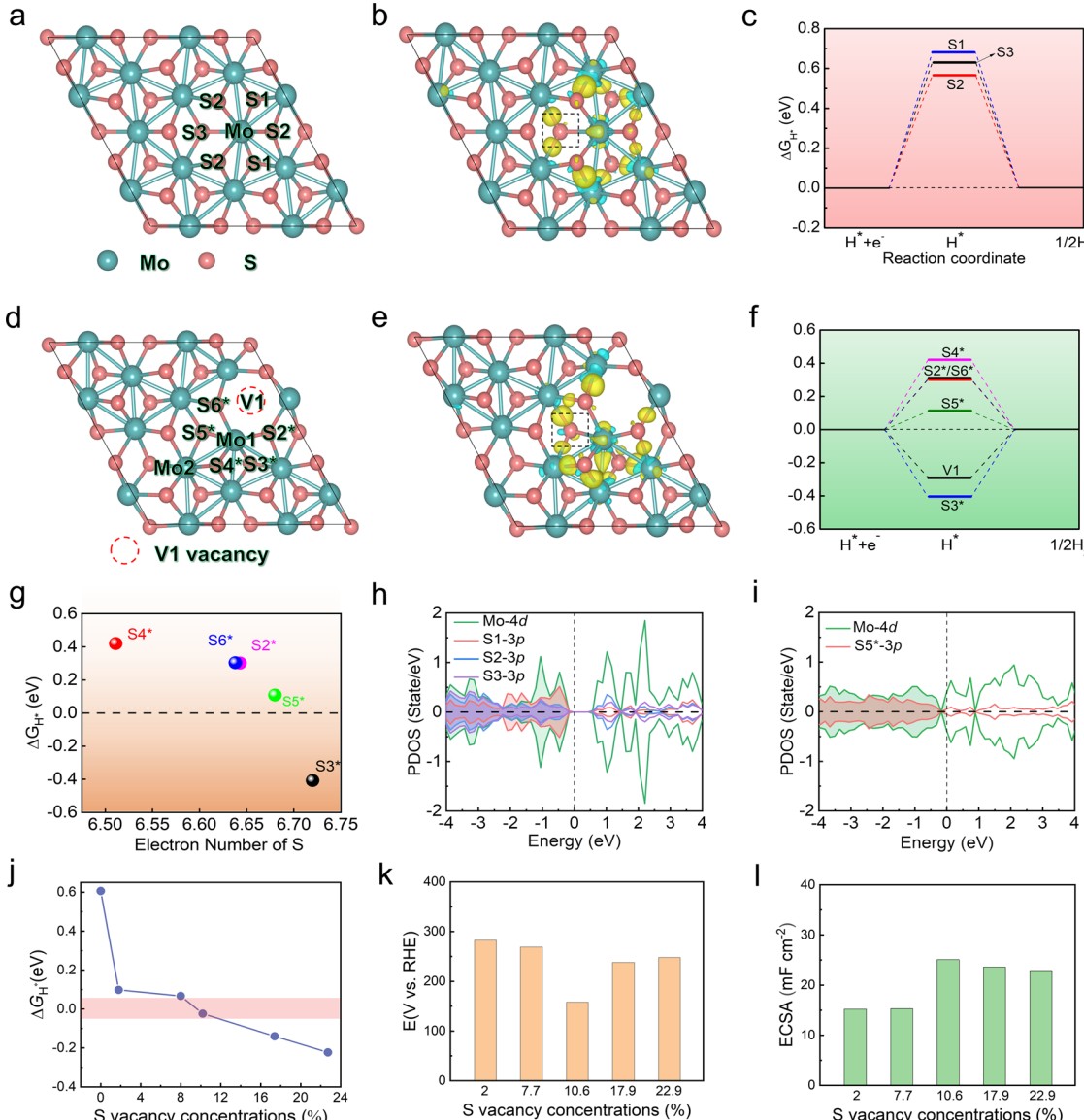

**Fig. 4 | Calculated hydrogen adsorption free energies and electronic structures of multiple active sites in 1T'''-MoS₂, 1T'''-MoS₂-V1 (V1: S vacancy). a, d** The optimized structure of 1T'''-MoS₂, 1T'''-MoS₂-V1. **b, e** The correspondingly calculated charge density difference of different adsorption S sites in 1T'''-MoS₂ and 1T'''-MoS₂-V1. The yellow and blue regions indicate the accumulation of positive and negative charge, respectively. **c, f** $\Delta G_{H^*}$ in different exposed S atoms around Mo in 1T'''-MoS₂ and 1T'''-MoS₂-V1. The $S_n/S_n^*$ ($n = 2–6$) corresponds to the labelled number of S atoms. **g** Correlation between $\Delta G_{H^*}$ and the electron numbers of S on 1T'''-MoS₂-V1. **h** Projected density of state (PDOS) of S1, S2 and S3 atom of 1T'''-MoS₂. **i** Projected density of state of S5* atom of 1T'''-MoS₂-V1. **j** The correlation between S vacancy concentrations and $\Delta G_{H^*}$. **k, l** Overpotential and ECSA of 1T'''-MoS₂ with different S vacancy concentrations.

continuous CV and chronoamperometry (CA) analysis (Fig. 3e and Supplementary Fig. 17). 1T'''-MoS₂–10.6% displays minor degradation (a small shift of about 11 mV at $j = 10$ mA cm⁻²) in the acidic solution. Meanwhile, CA analysis shows the HER current remains stable over a 24 h period with little degradation (the inset of Fig. 3e). No phase change or second phase is observed in the Raman spectra and XRD patterns before and after the electrochemical test (Fig. 3f and Supplementary Fig. 18), suggesting that the phase structure of 1T'''-MoS₂–10.6% maintains robust stability for HER. In previous studies, metallic monoclinic 1T'-MoS₂ (another metastable phase of MoS₂) with higher conductivity displays better HER performance than 2H-MoS₂-V_S but lacks electrochemical stability[52]. CA analysis shows the HER current of 1T'-MoS₂ is unstable over a 24 h period with significant degradation of 70% (Supplementary Fig. 19). Herein, 1T'''-MoS₂ shows both superior HER activity and long-term cycling stability. Compared with the recently reported nanostructured MoS₂ electrocatalysts, bulk 1T'''-

MoS₂–10.6% with smaller surface area has impressive electrochemical performance (Fig. 3g and Supplementary Table 5).

## Charge self-regulation effect on the HER activity

Based on the above analysis, we carried out the DFT calculation to further identify the charge self-regulation effect of active Mo−Mo bonds of 1T'''-MoS₂ in acidic conditions. Figure 4a and b show the relaxed structure and charge-density difference of multiple adsorption sites of 1T'''-MoS₂. Especially, there are several inherent Mo−Mo bonds and distorted octahedral coordination between S and Mo atoms (Fig. 4a). The calculated charge density difference among three different S adsorption sites is represented in Fig. 4b, indicating fewer electronic states of S atoms (yellow regions). The $\Delta G_{H^*}$ is an admissive descriptor for correlating the theoretical predictions with experimental measurements of catalytic activity[53]. The optimal value of $\Delta G_{H^*}$ close to zero, where adsorbed atomic hydrogen is in a thermo-neutral

state. The proton-coupled electron transfer can be accelerated, and then molecular hydrogen will be released[54]. As a reference, the calculated $\Delta G_{H^*}$ values on three different adsorption sites range from 0.567 to 0.682 eV (Fig. 4c), which are unfavorable for hydrogen adsorption.

As shown in Fig. 4d–f and Supplementary Fig. 20, three different vacancies and corresponding charge density difference in 1T'''-MoS$_2$ model were constructed to study the relationship between the hydrogen adsorption behavior and electronic structure (1T'''-MoS$_2$-Vn, $n = 1$–3). Particularly, S vacancies are introduced to activate Mo–Mo bonds and neighboring S atoms via the formation of dangling bonds (Fig. 4d). Accordingly, the electronic states of S atoms in 1T'''-MoS$_2$-V1 exhibit the charge redistribution surrounded by yellow regions in Fig. 4e. Compared to pristine 1T'''-MoS$_2$, the electronic states of S atoms increase due to the charge regulation effect of activated Mo–Mo bonds. Then, the S atoms connected with the activated Mo–Mo bonds exhibit the stronger bonding strength ranging from −0.407 to 0.420 eV (Fig. 4f). Therefore, the charge redistribution induced by the activated Mo–Mo bonds in 1T'''-MoS$_2$-V1 results in extra electrons of S atom around the V1 vacancy to promote the formation of S–H bond. $\Delta G_{H^*}$ of the optimal active S5* atom is 0.098 eV. The active electronic states of S5* atoms are strongly associated with the generation of Mo1-Mo2 bonds (Supplementary Fig. 20).

To further verify the charge regulation effect of Mo–Mo bonds for S atoms, the quantitative charge distribution on S atoms has been calculated (Fig. 4g). Bader charge analyses of S atoms on 1T'''-MoS$_2$-V1 revealed a linear correlation between the $\Delta G_{H^*}$ and the amount of electronic states of S atom. The activity of 1T'''-MoS$_2$-V1 is mainly attributed to the moderate amount of electrons of S5* atom, whereby the small amount of electrons of S2*, S4* and S6* atoms hardly facilitate the hydrogen adsorption while the large amount of electrons of S3* atom unfavorably promotes hydrogen desorption.

The enhanced HER activity is closely associated with not only the increased active sites but also the improvement of conductivity. The project of density of states of 1T'''-MoS$_2$ and 1T'''-MoS$_2$-V1 are further calculated to understand the intrinsic activity (Fig. 4h, i and Supplementary Fig. 21). Especially, the conductivity of pristine 1T'''-MoS$_2$ is relatively poor due to the large band gap (Fig. 4h). However, the 1T'''-MoS$_2$-V1 containing the activated Mo–Mo bonds is subjected to charge self-regulation for narrowing band gap and increasing number of new gap states across Fermi surface, which is favorable for the improvement of the conductivity (Fig. 4i). Thus, upon the introduction of S vacancies, Mo–Mo bond of 1T'''-MoS$_2$-V1 would induce charge engineering and result in self-regulation effect on active electronic states, thus enhancing conductivity and intrinsic catalytic activity. In addition, we have added related optimized structure of H adsorbed on 1T'''-MoS$_2$-V2/V3 systems (Supplementary Figs. 22 and 23). Meanwhile, to illustrate the compelling agreement between the theory and experiment, the trend in predicted $\Delta G_{H^*}$ as a function of vacancy concentration is compared with the trend in experimental performance metrics (overpotential, $R_{ct}$ and ECSAs). The optimal $\Delta G_{H^*}$ (−0.024 eV) of the S-vacancy model (Fig. 4j) is agreeable with the best performance of 1T'''-MoS2-10.6% (the lowest overpotential, minimum $R_{ct}$ and highest ECSA in Fig. 4k, l).

## Discussion

In summary, we have theoretically and experimentally revealed that the charge self-regulation effect in metastable 1T'''-MoS$_2$ could be utilized to manipulate the active electronic states and then optimize the HER activity. With the increase of S vacancies, the activated Mo–Mo bonds would further redistribute the active electronic states of neighboring S atoms to improve hydrogen adsorption. 1T'''-MoS$_2$-V$_S$ with different S vacancies was prepared and adjusted by a chemical etching method. The electrochemical results revealed that the optimal 1T'''-MoS$_2$–10.6% exhibited an overpotential of 158 mV at current density of 10 mA cm$^{-2}$ and a low Tafel slope of 74.5 mV dec$^{-1}$, far

superior to that of 2H-MoS$_2$-V$_S$, which verified the charge self-regulation effect of the activated Mo–Mo bonds on boosting the HER activities. This work provides a design concept of charge self-regulation effect and an efficient strategy of vacancy engineering for developing oxide/sulfide-based catalysts with superior performance for energy conversion and storage.

## Methods

### Synthesis of $K_{0.38}(H_2O)_yMoS_2$ precursor, 1T'-MoS$_2$, 1T'''-MoS$_2$−2.0%, 1T'''-MoS$_2$-Vs and 2H-MoS$_2$-V$_S$

The bulk KMoS$_2$ crystals were synthesized via high temperature solid-state reaction. 1 mmol of K$_2$S$_2$ powder, 1 mmol of Mo powder and 1 mmol of MoS$_2$ powder were mixed and pressed into a pellet. After sealed in carbon-coated fused silica tube, the mixture was transferred into muffle furnace and annealed at 1123 K for 20 h with the heating rate of 5 K min$^{-1}$. Afterwards, the samples were cooled down to room temperature naturally. Ultimately, the samples were washed with deionized water to obtain the $K_{0.38}(H_2O)_yMoS_2$ precursor and then dried in a vacuum oven at room temperature.

The $K_{0.38}(H_2O)_yMoS_2$ crystals were soaked in the 0.01 mol L$^{-1}$ H$_2$SO$_4$ aqueous solution of 0.004/0.05 mol L$^{-1}$ K$_2$Cr$_2$O$_7$ for about 15 min to harvest the 1T'-MoS$_2$ /1T'''-MoS$_2$-2.0%. The obtained product was washed with deionized water for several times and dried in a vacuum oven at room temperature.

1T'''-MoS$_2$−2.0% was immersed in the H$_2$SO$_4$ aqueous solution of 0.05 mol L$^{-1}$ K$_2$Cr$_2$O$_7$ at room temperature for varied durations (0.5 h, 1 h, 2 h, 3 h) to obtain 1T'''-MoS$_2$-V$_S$ (V$_S$ = 7.7%, 10.6%, 17.9%, 22.9%). 1T'''-MoS$_2$-V$_S$ was washed with deionized water several times and dried in a vacuum oven at room temperature.

The defective 2H-MoS$_2$ sample was prepared by annealing of 1T'''-MoS$_2$-V$_S$ in an evacuated quartz tube at 513 K for 5 h. The reference 2H-MoS$_2$-V$_S$ for comparison was derived from 1T'''-MoS$_2$−10.6%.

### Structural characterization

X-ray diffraction (XRD) measurement was carried out by Bruker D8 advance. Raman spectra were obtained using a thermal dispersive spectrometer with laser excitation at 633 nm and the laser intensity is set as low as 5% to avoid the phase transition of metastable 1T'''-MoS$_2$ during the test. Material microstructure was characterized by field emission scanning electron microscope (SEM, Hitachi S-4800), transmission electron microscopy (TEM, JEOL JEM-2100F) and spherical aberration-corrected transmission electron microscope (ACTEM, Hitachi HF5000). The images were obtained in high angle annular dark field (HAADF) and bright field (BF) to harvest the mass-thickness and diffraction contrast information, respectively. The internal and external receiving angles of HAADF imaging were 68 and 280 m rad, respectively, and the receiving angles of BF imaging were 17 m rad. X-ray photoelectron spectroscopy (XPS) was measured in a Thermo VG Scientific with Al Kα radiation (λ = 1486.6 eV). The X-ray absorption spectra (XAS) including X-ray absorption near-edge structure (XANES) and extended X-ray absorption fine structure (EXAFS) of the sample at Mo K-edge and Mo L$_3$-edge were collected at the Beamline of TLS07A1 in National Synchrotron Radiation Research Center, Taiwan. Total K and Mo contents were quantified by an inductively coupled plasma optical emission spectrometer (ICP–OES, Agilent 5110, Australia).

### Electrochemical measurements

The electrochemical experiment was carried out using a CHI 760E electrochemical workstation at ambient temperature. The measurements were performed in 0.5 mol L$^{-1}$ H$_2$SO$_4$ solution (deaerated by Ar) using a three-electrode setup, with a saturated calomel electrode (in saturated KCl solution) reference electrode, a graphite rod counter electrode and the glassy carbon working electrode. The catalyst was prepared by ultrasonically dispersing in Nafion/alcohol solution (0.5 wt.%, Alfa Aesar) to obtain 5.0 mg mL$^{-1}$ slurry. The catalyst was

dispersed in Nafion/alcohol solution (0.5 wt.%, Alfa Aesar) by sonication for 60 min in ice water. Then, 10 μL of the mixed solution was drop-casted onto a glassy carbon rotating disk working electrode (5 mm diameter) and dried with $N_2$. Initially, cyclic voltammogram (CV) was operated at least 20 cycles to guarantee the activation of catalyst and then linear sweep voltammetry (LSV) was performed at a scan rate of 0.01 V s$^{-1}$ in the potential range of 0.1 ~ −0.36 (vs. RHE). The ECSA of catalyst was tested as the double-layer capacitance ($C_{dl}$) in the potential window of −0.145 ~ −0.045 V (vs. RHE) at the scan rates of 20, 40, 60, 80, 100, and 120 mV s$^{-1}$ in the CV model. The calibration of SCE reference electrode is performed in a standard three-electrode system with Pt wires as the working electrode, graphite rod as the counter electrode, and the SCE as the reference electrode. Electrolytes are pre-purged and saturated with high purity $H_2$. CVs were run at a scan rate of 1 mV s$^{-1}$, and the average of the two potentials at which the current crossed zero was taken to be the thermodynamic potential for the HER (Supplementary Fig. 24). All the potentials were converted to the potential vs. the reversible hydrogen electrode (RHE). In 0.5 M $H_2SO_4$, E(RHE) = E(SCE) + 0.272 V. The rotating disk working electrode was rotated at 1600 rpm to remove the hydrogen gas bubbles formed at the catalyst surface. The current vs. potential plots were corrected by 90% ohmic compensation. In addition, the constant current (10 mA cm$^{-2}$) measurements were also implemented to evaluate the stability of potential.

In order to avoid the influence of the above factors on HER activity, the catalysts were dispersed in Nafion/alcohol solution (0.5 wt. %, Alfa Aesar) by sonication for 60 min in ice water. Then, 10 μL of the mixed solution was drop-casted onto a glassy carbon rotating disk working electrode (5 mm diameter) and dried with $N_2$. Moreover, three electrodes were prepared to test HER testing, which are labeled as electrode 1, electrode 2 and electrode 3, respectively. LSV curves are shown in Supplementary Fig. 25. The average values and standard deviation data of the overpotential of three electrodes tests have been added in Supplementary Fig. 26 and Supplementary Table 6. The average values of 1T'''-MoS$_2$–2.0%, 1T'''-MoS$_2$–7.7%, 1T'''-MoS$_2$–10.6%, 1T'''-MoS$_2$–17.9%, 1T'''-MoS$_2$–22.9%and 2H-MoS$_2$-V$_S$ are 285.33, 266.33, 158.33, 239.00, 246.00 and 364.67 mV, respectively. The standard deviation values of 1T'''-MoS$_2$–2.0%, 1T'''-MoS$_2$–7.7%, 1T'''-MoS$_2$–10.6%, 1T'''-MoS$_2$–17.9%, 1T'''-MoS$_2$–22.9%and 2H-MoS$_2$-V$_S$ are 2.62, 3.09, 1.25, 2.16, 2.16 and 4.78 mV, respectively.

### Computational methods

The Vienna Ab initio Simulation Package was used to perform spin-polarized density functional theory (DFT) calculations, and the generalized gradient approximation of Perdew-Burke-Ernzerhof was introduced to describe electron exchange and correlation[55,56]. The cut-off of plane-wave basis was set as 450 eV[38]. The projector-augmented plane wave was used to describe the electron-ion interacitons[57]. The em piracal dispersions of Grimme (DFT-D2) was applied to account for long-range van Waals interactions[58]. A set of (4 × 4 × 1) k-points were selected for geometric optimization, and the convergence threshold was set as 10$^{-4}$ eV in energy and 0.05 eV/Å in force, respectively. In the electronic structure calculation, denser k-points (8 × 8 × 1) were used for better accuracy. The vacuum slab of 15 Å was inserted in the z-direction for surface isolation to elimate periodic interaction. The structure of 1T'''-MoS$_2$ containing 36 atoms was introduced to model a system. The free energy of the adorbed state was calculated as

$$\Delta G = \Delta E_{H^*} + \Delta E_{ZPE} - T\Delta S \qquad (1)$$

where $\Delta E_{H^*}$ is the hydrogen chemisorption energy, and $\Delta E_{ZPE}$ is the difference of the zero point energy between the adsorbed state and the gas phase. Considering the fact that the vibriation entropy of H* in the adsorbed state is very small, the entropy of 1/2 $H_2$ adsorption can be approximated as $\Delta S_H \approx -1/2 S_{H^2}^0$, where $S_{H^2}^0$ is the entropy of $H_2$ in the gas phase at the standard conditions.

## Data availability

All data supporting the findings in the paper as well as the Supplementary Information files are available from the corresponding authors on reasonable request.

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

## Acknowledgements

This study was supported by the National Natural Science Foundation of China (Grant No. 21871008, 21801247, 21872166, 21973107 and 51702345), the Science and Technology Commission of Shanghai Municipality (21ZR1473300, 21ZR1472900, 22ZR1471600) and the Key Research Program of Frontier Science, Chinese Academy of Sciences (Grant No. QYZDJ-SSW-JSC013).

## Author contributions

F.H., J.L., and Wei.Z. conceived the research. X.G. and Y.F. carried out the synthesis. X.G. performed materials characterization and electro-chemical measurements. Wen.Z. and Y.L. conducted ACTEM measure-ment and analyzed data. E.S. proposed the active electronic states and carried out the theoretical calculations. X.G., E.S., Wei.Z., and S.X. drafted the paper. X.G., E.S., and Wei.Z. contributed equally to this work. All authors discussed the results and commented on the paper.

## Competing interests

The authors declare no competing interests.
