## [Peer Review File · Nature Communications]

Charge self-regulation in 1T''-MoS₂ structure with rich S vacancies for enhanced hydrogen evolution activityREVIEWER COMMENTS

Reviewer #1 (Remarks to the Author):

In this manuscript, the authors claimed that they synthesized the 1T^{'''}-MoS₂-VS catalyst and obtained high HER property. After carefully checked both the manuscript and supplementary information, I do not recommend the publication of this paper in Nature Communications. In addition, I strongly suggest the authors carefully double-check their raw data. Supplementary Figures 13a and f show the same noise models for different materials, which raises serious concerns and prevents publication in any newspaper. Nor have the authors clearly demonstrated the novelty of the 1T^{'''} phase compared to the well-studied 1T' phase (not even mentioned in the manuscript!). For example, the authors strongly argued that the 2H phase is semiconducting, which is not desirable in electrocatalysis. However, the authors report that the 1T^{'''} phase is semiconducting with a band gap of 0.65 eV and significantly greater than that of the phase 1T' of MoS₂. A systematic and careful evaluation of the performance of the 1T^{'''} phase versus that of the 1T' phase is also missing. Additional clarification throughout the manuscript is also welcome as indicated below.

Please see additional specific comments below.

1. The authors mentioned that Supplementary Figure 13a and supplementary figure 13f present the double layer capacitance for different materials. Please explain why they have totally the same curve (even the same noise pattern) albeit a different y-axis? As I know this is impossible to have the same noise for that such a large potential window. In Figure S3f and Figure S3a also show the same noise pattern and are identical.
2. What is the atomic structure for 1T^{'''} MoS₂, I suggest the author show the top and side views of the atomic structure in the Supplementary Information and provide the .cif data.
3. I suggest the author extend the discussion about the phase transition mechanism, why the K⁺ insertion can result in 1T^{'''} MoS₂? Why not 1T' or 1T^{''}. The SAED of the 3 respective phases should be provided and compared.
4. Please enlarge the x axis of the XPS spectra from the Mo and S regions to confirm the absence of contributions from oxidized Mo and S (Mo-ox and S-ox).
5. In Figure 2b, What is the intensity of the laser for the Raman spectrum?
6. In Figure 2d and 2c, I suggest the author compare with 2H MoS₂ to clarify the phase transition.
7. Mo L3 edge can also help to distinguish the phase transition, please provide.
8. Figure 2f and Figure 2g the dpi is not clear enough for me to recognize the S vacancies. Could the author provide a clearer image? The axis for Figure 2g is important, please make it more clear for the reviewers to recognize. Why does the author use ACTEM not HAADF-STEM to identify the S vacancy?
9. Figure 2e looks more like an FFT refined HRTEM / HAADF-STEM, please show the raw HAADF-STEM image here in case of distortion.
10. With the increasing percentage of phase transition, the Tafel slope is not changing a lot while the onset potential is dramatically changing, can the author try to explain why?
11. The medium percentage of 1T^{'''} phase of MoS₂ has the best EIS. The authors should comment on that.
12. Figure 3g should use double Y to better illustrate the difference between Tafel slopes.
13. The author should also compare the K-weighted EXAFS with 2H MoS₂.

Reviewer #2 (Remarks to the Author):

Guo et al. present a study of vacancy-rich 1T^{'''}-MoS₂ as an HER catalyst, showing that the introduction of controlled vacancy concentrations can tune the electronic structure and hence the conductivity and HER activity. Overall, the conclusions appear to be well supported by the data, and

the performance is impressive. The topic is also of broad interest to the community. I have a few comments:

1) Although the experimental methods are described in the Methods section, the computational methods are not. The authors should clearly indicate factors such as DFT functional, energy cutoffs, etc., which can affect the predicted degree of charge localization and transfer. Likewise, the details of how ΔG^* was computed are missing.

2) The concept of self-regulation of charge, which the authors use in their title, is still a bit nebulous to me. Does this just mean tunability of electronic structure? In what sense is the material "self-regulating"? The term implies a capability to dynamically reorganize charge under different conditions (i.e., upon H adsorption), but I don't see that reflected. This should be clarified.

3) In many instances, 2D materials can exhibit vacancy clustering. In general, clustering is difficult to diagnose using the experimental techniques used by the authors, but it can be easily investigated theoretically. The models used by the authors assume isolated vacancies, but I would be interested to see whether the performance benefits are predicted to change at all upon formation of a divacancy or larger vacancy cluster. This could be related to the slight degradation in performance observed upon cycling, for instance.

4) The agreement between the theory and experiment for the optimal vacancy concentration is compelling. Although it's largely shown in the SI, I would recommend including in the main text a plot that directly compares the trend in predicted ΔG_{H^*} as a function of vacancy concentration with the trend in an experimental performance metric (e.g., Tafel slope, Z_{ct} , or overpotential @ -10 mA/cm^2) as a function of vacancy concentration. In principle, the calculations can be used to estimate the precise vacancy concentration that would lead to $\Delta G_{H^*}=0$, which could act as a guideline for optimizing future synthesis.

5) The calculations find one site in particular ($S5^*$ in Fig. 4) that becomes active upon vacancy incorporation. This suggests a 1:1 correspondence between vacancy concentration and active site density. It would be interesting to see a comparison of active site density (extracted from the capacitance measurement of the electrochemically active surface area) versus the vacancy density to see how well this relationship holds up.

Reviewer #3 (Remarks to the Author):

In this manuscript, the authors demonstrated a wet chemical etching method using $K_2Cr_2O_7$ to induce S vacancies in the $1T''''-MoS_2$, which could result in the enhanced HER activity compared with the $2H-MoS_2$ with S vacancy. It was found that the generated S vacancies would activate the Mo-Mo bonds in $1T''''-MoS_2$ to redistribute the electronic states of neighboring S atoms. The $1T''''-MoS_2-10.6\%$ catalyst showed the best HER activity among all the tested samples, with an overpotential of 158 mV at the current density of 10 mA cm^{-2} and a Tafel slope of 74.5 mV dec^{-1} . Additionally, the mechanism of the enhanced HER activity was comprehensively studied and confirmed by the DFT calculations, supporting the authors' conclusions. Although this method is promising and may provide a new way for rational design and preparation of cost-effective and high-performance electrocatalyst, there are still several issues needed to be solved before this manuscript can be published.

1. It is noticed that the authors performed 60-min sonication of the catalyst solution before carrying out all the electrochemical tests. How can the metastable $1T''''-MoS_2-VS$ catalyst maintain the phase during such long-time sonication? Whether the $1T''''-MoS_2-VS$ is stable than the pure $1T''''-MoS_2$? The authors need to provide more experimental evidence to explain it.

2. It is doubtful that the reference $2H-MoS_2-Vs$ sample was obtained by thermal treatment of $1T''''-MoS_2-10.6\%$, how can the authors be sure that all the S vacancies would keep the same (amount and

positions) as that in the original 1T''-MoS₂-10.6% sample?

3. The authors used the strong oxidating agent to treat the samples to achieve the 1T''-MoS₂-VS catalysts with S vacancies up to 22.9%. It is highly possible that oxidation would happen simultaneously. However, characterization of oxidation is absent in this paper. The authors need to use XPS, EDS, and other related characterizations to reveal it. If the samples were not oxidized, they need to explain the reason why only the S vacancies are introduced using this method.

4. In supplementary figure 4, 1T''-MoS₂-10.6% sheet shown in the TEM image seems not single crystal, but the SAED showed clear spot diffraction pattern from a single crystal, please check this data.

5. In supplementary figure 5, the authors showed the clear XRD peak shift between the 1T''-MoS₂-samples with different percentage of S vacancies. What is the origin for this peak shift? It is hard to believe that small percentage of S vacancies can induce this peak shift. It may be caused by some oxidation or phase transition during the treating process. It would be better the authors can also show the Raman spectra of all the 1T''-MoS₂-Vs samples.

6. In supplementary figure 10, the DFT data show the results of samples with V_s=0%, 2.8%, 9.1%, 12.5%, 16.1%, which are different with the real data obtained in the prepared samples. Why not perform the calculations based on the same vacancy values to build models? If so, it will make the authors' conclusions more robust and convincing.

7. Another issue for the DFT calculation model is in Figure 4a, there are 3 S₂, 2 S₁ and 1 S₃ sites in the draw scheme. Does that mean there are 3 types of S sites in the 1T''-MoS₂ structure, represented by S₁, S₂ and S₃? Why the numbers are different?

8. It seems that the concentration of S vacancy in the reported samples can be as high as 22.9%, can these samples maintain their structure during the test? What is the stability for all the 1T''-MoS₂-Vs samples with 2.0%, 7.7%, 10.6%, 17.9%, 22.9% S vacancies?

9. The concentration of S vacancy is calculated based on the XPS results by converting the ratio of S and Mo. How did the authors exclude the existence of Mo vacancies? Moreover, there should be various types of S vacancies in the 1T''-MoS₂-Vs samples, how to identify them, and which kind of S vacancy has the best HER activity?

10. In Figure 2e and f, the HAADF-STEM image in Figure 2e is clearer than the Figure 2f, which is more suitable for the characterizations of S vacancies. The data obtained from Figure 2f are not convincing.

11. For the electrochemical test, the authors should also compare their results with pristine 1T''-MoS₂ and 2H-MoS₂ samples to make the conclusions more reliable and convincing.

Point-to-point Responses

Reviewer: 1

Overall comment

In this manuscript, the authors claimed that they synthesized the 1T'''-MoS₂-V_S catalyst and obtained high HER property. After carefully checked both the manuscript and supplementary information, I do not recommend the publication of this paper in Nature Communications. In addition, I strongly suggest the authors carefully double-check their raw data. Supplementary Figures 13a and f show the same noise models for different materials, which raises serious concerns and prevents publication in any newspaper. Nor have the authors clearly demonstrated the novelty of the 1T''' phase compared to the well-studied 1T' phase (not even mentioned in the manuscript!). For example, the authors strongly argued that the 2H phase is semiconducting, which is not desirable in electrocatalysis. However, the authors report that the 1T''' phase is semiconducting with a band gap of 0.65 eV and significantly greater than that of the phase 1T' of MoS₂. A systematic and careful evaluation of the performance of the 1T''' phase versus that of the 1T' phase is also missing. Additional clarification throughout the manuscript is also welcome as indicated below.

Response: We sincerely appreciate the Reviewer to give constructive comments, and the questions raised are helpful to improve our paper's quality and depth.

- (1) The raw data have been carefully examined. We apologize for our careless mistake when processing the data, and truly appreciate your guidance. The correct data were provided, and the corresponding curve was plotted.
- (2) 1T'''-MoS₂ is considered to be a more stable phase than 1T'-MoS₂, and both of them are derived from metastable 1T-MoS₂ due to Peierls distortions (*Phys. Rev. B.* 2016, **94**, 195402). 1T'-MoS₂ is metallic phase ($\sigma = 618 \text{ S cm}^{-1}$) and 1T'''-MoS₂ is semiconductor phase ($E_g = 0.65 \text{ eV}$, $\sigma = 2.26 \text{ S cm}^{-1}$; *Phys. Rev. B.* 2018, **98**, 184513; *J. Am. Chem. Soc.* 2019, **141**, 790–793), which are more conductive than 2H-MoS₂ ($E_g = 1.20 \text{ eV}$, $\sigma = \sim 10^{-4} \text{ S cm}^{-1}$). Although the HER electrocatalytic activities of

1T'-MoS₂ reported in the recent literature are superior to 2H-MoS₂, the chemical and electrochemical stability is poor (*Nat. Chem.* 2018, **10**, 638–643; *Chem. Commun.* 2018, **54**, 12714–12717). Phase transition from 1T'-MoS₂ to 2H-MoS₂ was observed less than one week in air at room temperature, while 1T'''-MoS₂ is still stable more than one month (**Figure R1a, b**). The Raman spectra show four characteristic peaks of 1T'-MoS₂ located at 150 (*J*₁), 212(*J*₂), 280 (*E*_{1g}) and 324 (*J*₃) cm⁻¹ (*Phys. Rev. B*, 2018, 98, 184513). The main peaks at 15.08°, 31.34° and 34.35° in the X-ray diffraction (XRD) pattern are assigned to the (200), (-202) and (002) facets of the 1T'-MoS₂ (**Figure R1c**).

The HER electrocatalytic results of 1T'-MoS₂ are included in the revision which could be better to convince the researchers outside this specific field (**Figure R1d-f**). 1T'-MoS₂ exhibits an overpotential of 240 mV at current density of 10 mA cm⁻² and a Tafel slope of 85.0 mV dec⁻¹. Chronoamperometry (CA) analysis shows the HER current is unstable over a 24 h period with significant degradation (70%). During the HER process, 1T'-MoS₂ is unstable, and is gradually converted to 2H-MoS₂. Compared with 1T'-MoS₂, our 1T'''-MoS₂ can maintain the stability of phase structure after 24-hour cycling test. To clarify more clearly, the phase-selective synthesis of metastable 1T' and 1T''' polymorphs and the detailed comparison of their HER performance are supplemented in the revised manuscript. The HER performance is compared between our 1T'''-MoS₂ and 1T'-MoS₂ based and 2H-MoS₂-based electrocatalysts in **Table R1** and **Supplementary Table 5**.

- (3) It is the first time that the HER performance of 1T'''-MoS₂ is investigated and defect engineering is adopted. We focus on proposing a novel concept of charge self-regulation to manipulate the active electronic states of transition metal dichalcogenide (TMD) and present a simple method to create S-vacancies in metastable MoS₂ to boost electrocatalytic activity. As a proof-of-concept, metastable 1T'''-MoS₂ with featured Mo–Mo bonds is studied as a novel electrocatalyst for HER from both computational and experimental investigations. The rich Mo–Mo bonds in 1T'''-MoS₂ serve as an electron reservoir to effectively facilitate and manipulate the electronic state of S atoms. Charge engineering

induced by Mo–Mo bonds contributes to a self-regulation effect on manipulating the active electronic states after S vacancy generation, which regulates hydrogen adsorption free energy close to thermoneutral and then promotes the electrochemical performance. A facile chemical etching strategy is used to introduce and control S-vacancy concentrations in 1T^{'''}-MoS₂ bulk crystal. After optimization, 1T^{'''}-MoS₂ delivers a low overpotential of 158 mV at 10 mA cm⁻² and Tafel slope of 74.5 mV dec⁻¹, far exceeding that of 2H-MoS₂ (369 mV and 137 mV dec⁻¹). In addition, we give a full discussion on the influence of S-vacancy on the HER activity by combining experimental analysis and DFT calculations. This work provides novel design and synthetic strategy for developing new class of highly active TMD-based electrocatalysts.

We sincerely hope that our elaborative revision and full discussions could meet the requirements of the Reviewer. We sincerely expect this work to be published in this journal, which may be beneficial to more researchers.

Figure R1. Raman spectra of (a) 1T[']-MoS₂ before and after one week of storage and (b) 1T^{'''}-MoS₂ before and after one month of storage. E_{2g} and A_{1g} (yellow dotted line frame) belong to 2H-MoS₂. (c) XRD pattern of the 1T[']-MoS₂. HER performance of the 1T[']-MoS₂ sheet (d) linear sweep polarization curve, (e) Tafel curve and (f) the long-term durability test.

Table R1. Comparison of the HER performance of the 1T'''-MoS₂ with recently-reported 1T/1T'-MoS₂ catalysts.

Catalysts	MoS ₂ phase	Overpotential (mV)	Tafel slope (mV dec ⁻¹)	Reference
1T'''-MoS ₂	1T'''	158	74.5	This Work
1T'-MoS ₂	1T'	240	85.0	This Work
1T'-MoS ₂	1T'	201	100	Nat. Chem. 2018, 10 , 638–643
Ni@1T-MoS ₂	1T/2H	218	/	Nat. Comm. 2020, 11 , 4114
1T-MoS ₂ nanosheet	1T	230	45	ACS Energy Lett. 2018, 3 , 7-13
1T _{0.72} -MoS ₂ @NiS ₂	1T/2H	138	42	Nat. Comm. 2021. 12 , 5260
Co-BDC/MoS ₂	1T/2H	248	109	Small. 2019, 15 , 1805511
1T-MoS ₂	1T/2H	271	61	J. Am. Chem. Soc. 2018, 140 , 441-450
1T-MoS ₂ /NiS ₂	1T/2H	116	72	Angew. Chem. Int. Ed. 2019, 58 , 17621–17624
MoS ₂ -GNR	1T/2H	205	50	Adv. Funct. Mater. 2018, 1802744
(N,PO ₄ ³⁻)-MoS ₂ /VG	1T/2H	85	42	Angew. Chem. Int. Ed. 2019, 58 , 16289–16296
1T'-MoS ₂	1T'	300	83	Adv. Mater. 2017, 29 , 1701955
T-MoS ₂	1T/2H	290	78	Nat. Energy. 2017, 2 , 17127

Comment 1. The authors mentioned that Supplementary Figure 13a and supplementary figure 13f present the double layer capacitance for different materials. Please explain why they have totally the same curve (even the same noise pattern) albeit a different y-axis? As I know this is impossible to have the same noise for that such a large potential window. In Figure S3f and Figure S3a also show the same noise pattern and are identical.

Response: We appreciate the Reviewer for pointing out this low-level mistake. The raw data related to cyclic voltammetry (CV) curves of both 1T'''-MoS₂-2.0% and 2H-MoS₂-V_s are carefully double-checked. **Supplementary Figure 16a** is the CV of 1T'''-MoS₂-2.0%. The correct data of 2H-MoS₂-V_s were provided, and the corresponding

curve was plotted (**Figure R2**).

Figure R2. (a, c) CVs of 1T''-MoS₂-2.0% and 2H-MoS₂-V_S (b, d) Plots of current density versus scan rate.

Change to the Manuscript (Page 12): Electrochemical active surface areas (ECSAs) are estimated from the double-layer capacitance (C_{dl}) by measuring cyclic voltammetry (CV) curves. According to **Figure 3c** and **Supplementary Figure 16**, 1T''-MoS₂-10.6% possesses the highest value of ECSA (25.1 mF cm⁻²), considerably larger than those of 1T''-MoS₂-2.0% (15.2 mF cm⁻²), 1T''-MoS₂-7.7% (15.3 mF cm⁻²), 1T''-MoS₂-17.9% (23.6 mF cm⁻²), 1T''-MoS₂-22.9% (22.9 mF cm⁻²), commercial 2H-MoS₂ (1.17 mF cm⁻²) and 2H-MoS₂-V_S (0.97 mF cm⁻²), suggesting proper concentration of S vacancies can significantly expand the ECSAs and expose more electrochemically active sites.

Comment 2. What is the atomic structure for 1T'' MoS₂, I suggest the author show the top and side views of the atomic structure in the Supplementary Information and provide the .cif data.

Response: Thanks for the Reviewer's suggestion. The top and side views of the atomic structure have been shown in **Figure R3**. Single-crystal X-ray diffraction is used to

determine the crystal structure of 1T^{'''}-MoS₂, which crystallizes in the trigonal space group P31m with lattice parameters, $a = 5.580 \text{ \AA}$ and $c = 5.957 \text{ \AA}$, as listed in **Supplementary Table 1-3**. The Mo–S bond lengths are 2.370 \AA , 2.521 \AA , 2.459 \AA , and 2.298 \AA , compared with 2.413 \AA in 2H-MoS₂, 2.384 \AA in 3R-MoS₂, 2.385 \AA , 2.394 \AA , 2.459 \AA and 2.507 \AA in 1T'-MoS₂ and 2.389 \AA in 1T-MoS₂. These equivalent Mo–Mo bonds (3.015 \AA) in 1T^{'''}-MoS₂ form corner-sharing Mo₃ trimers in the ab plane, compared with the Mo–Mo zigzag chains (Mo–Mo bond, 2.774 \AA) in 1T'-MoS₂. Moreover, the nearest Mo atoms form the Mo–Mo trimers, in which the Mo–Mo bond length (3.013 \AA , measured in aberration-corrected HAADF-STEM image, **Figure 2e**) is consistent with the value from the single crystal X-ray diffraction data. The above results are consistent with the calculated model and our reported crystal structure data (*J. Am. Chem. Soc.* 2019, **141**, 790–793).

Figure R3. The top view and side view of 1T^{'''}-MoS₂.

Change to the Manuscript (Page 6): Crystal structure data of 1T^{'''}-MoS₂ were measured by the single crystal X-ray diffraction. The top and side views of the atomic structure of 1T^{'''}-MoS₂ have been shown in the **Supplementary Figure 1**.

Comment 3. I suggest the author extend the discussion about the phase transition mechanism, why the K⁺ insertion can result in 1T^{'''} MoS₂? Why not 1T' or 1T". The SAED of the 3 respective phases should be provided and compared.

Response: Thanks for your comments. Metastable polymorphs of MoS₂ (1T, 1T', 1T^{'''}) are derived from alkali metal intercalated compounds of MoS₂. In this work, K_x(H₂O)_yMoS₂ used as a precursor crystallizes in the monoclinic structure with featured zig–zag chains of Mo–Mo bonds. The K⁺ removal from K_x(H₂O)_yMoS₂ results in 1T'

and 1T^{'''}-MoS₂. In the process of removing K⁺ from the precursor, the Mo–Mo bonds are re-arranged to cause different phase structures at the different mass concentration of oxidant (K₂Cr₂O₇). With the increase of concentration to 0.05 mol L⁻¹, the phase structure gradually changes from 1T' to 1T^{'''}, as shown in **Figure R4**. The SAED patterns (**Figure R5**) also confirm the crystal structures of the 2H (P6₃/mmc), 1T' (C2/m) and 1T^{'''} phase (P31m). The crystal structure information and selected bond distances (Å) for 1T'-MoS₂, 1T^{'''}-MoS₂ and 2H-MoS₂ are shown in **Tables R2** and **R3**.

The 1T'-MoS₂ is obtained following the topotactic reaction mechanism and it inherits the monoclinic structure and Mo–Mo zigzag chains of the parent compound K_x(H₂O)_yMoS₂. However, the crystal structure of 1T^{'''}-MoS₂ is completely re-arranged due to the depotassiation reaction in a stronger oxidation environment. Therefore, as long as the content of oxidant in the soft chemical reaction process is controlled, phase-selective synthesis of 1T' and 1T^{'''} polymorphs can be realized. We also attempted to synthesize the theoretically predicted 1T'' phase (*Phys. Rev. B.* 2017, **96**, 195426; *Phys. Rev. B.* 2017, **96**, 165305), but failed to obtain large-size (> 20 μm) samples for single crystal refinement.

Figure R4. (a) XRD patterns of various metastable MoS₂ samples obtained using different concentrations of oxidizing agent K₂Cr₂O₇ (0, 0.004, 0.01, 0.02, 0.03, 0.04 and 0.05 mol L⁻¹).

Figure R5. SAED patterns of (a) 2H, (b) 1T', and (c) 1T'''-MoS₂.

Table R2. Crystal structure information for 1T'-MoS₂, 1T'''-MoS₂ and 2H-MoS₂.

	1T'	1T'''	2H
Crystal structure	monoclinic	trigonal	hexagonal
Space group	C2/m	P3 ₁ m	P6 ₃ /mmc
Unit cell	a = 12.84 Å b = 3.205 Å c = 5.709 Å	a = 5.580 Å b = 5.580 Å c = 5.957 Å	a = 3.161 Å b = 3.161 Å c = 12.298 Å
Volume	214.95 Å ³	160.63 Å ³	106.4 Å ³
[MoS ₆] coordination	distorted octahedron	distorted octahedron	trigonal prism
Mo–Mo distance	2.803 Å 3.205 Å 3.775 Å	3.015 Å	3.161 Å
Mo–Mo bonding	Yes Zigzag chain	Yes Diamond chain	No
Electronic property	Superconductor	Semiconductor	Semiconductor

Table R3. Selected bond distances (Å) for 1T' MoS₂, 1T'''-MoS₂ and 2H-MoS₂.

	1T'-MoS ₂	1T'''-MoS ₂	2H-MoS ₂
Mo–Mo	2.803 3.205 3.775	3.015	3.161
Mo–S	2.385 2.394 2.459 2.507	2.298 2.371 2.459 2.522	2.413

Comment 4. Please enlarge the x axis of the XPS spectra from the Mo and S regions

to confirm the absence of contributions from oxidized Mo and S (Mo-ox and S-ox).

Response: We thank the Reviewer's valuable comments. The x axis of the XPS spectra is enlarged from 224 ~235 eV (160 ~166 eV) to 223 ~238 eV (158 ~168 eV) (**Figure R6**), which show that Mo remains in +4 valence state and S in -2 valence state. No peaks assigned to the oxidized Mo and S (Mo-ox and S-ox) species appear.

Figure R6. Mo 3d (a) and S 2s (b) XPS spectra of 1T''-MoS₂-10.6%.

Comment 5. In Figure 2b, what is the intensity of the laser for the Raman spectrum?

Response: Raman spectra were obtained using a thermal dispersive spectrometer with laser excitation at 633 nm and the laser intensity is set as low as 5% to avoid the phase transition of metastable 1T''-MoS₂ during the test.

Comment 6. In Figure 2d and 2c, I suggest the author compare with 2H MoS₂ to clarify the phase transition.

Response: Thanks for your suggestion. The XAFS data of 2H-MoS₂ sample is added (**Figure R7**) for comparison.

Figure R7. (a) FT-EXAFS spectra and (b) Mo K-edge XANES spectra of Mo foil, 1T''-MoS₂-2.0%, 1T''-MoS₂-10.6% and 2H-MoS₂.

Change to the Manuscript (Page 8): The local bond lengths and coordination environment of 1T''-MoS₂-V_S (V_S = 2.0%, 10.6%) were studied by X-ray absorption spectroscopy (XAS) with Mo foil and commercial 2H-MoS₂ as reference. The Mo K-edge X-ray adsorption near edge structure (XANES) spectra (**Figure 2c**) indicate that 1T''-MoS₂-2.0% and 1T''-MoS₂-10.6% possess similar absorption edge and white line peaks, but different from 2H-MoS₂ and Mo foil. The oxidation state of Mo element in Mo foil is zero while in 2H-MoS₂ is +4. **Figure 2d** shows the result of Fourier transform of the extended X-ray absorption fine structure (FT-EXAFS) oscillation. The two characteristic peaks at 1.9 and 2.8 Å (not corrected by scattering phase shift) can be assigned to the nearest Mo–S and Mo–Mo bonds, respectively.⁴⁵ No obvious shift is observed in 1T''-MoS₂-10.6% with higher S vacancy concentration, indicating Mo–Mo bond length is nearly unchanged before and after chemical etching. The characteristic peak (2.9 Å) of 2H-MoS₂ can be attributed to longer Mo–Mo distance. However, the reduced peak intensity of Mo–S bonds and unchanged peak intensity of Mo–Mo bonds in 1T''-MoS₂-10.6% manifests the decrease of coordination number, probably due to the generation of more defects after increasing S vacancies. More S defects in 1T''-MoS₂-10.6% result in a decrease in Mo K-edge oscillation functions $k^3 \chi(k)$ at the k range of 0–14 Å⁻¹ (**Supplementary Figure 8**) compared with 1T''-MoS₂-2.0% and 2H-MoS₂. Furthermore, the Mo L₃-edge XANES spectra of 2H-MoS₂ and 1T''-MoS₂-10.6% were compared. For 2H-MoS₂, the Mo L₃-edge white line peak locates at 2524.01 eV

and it is ascribed to the Mo^{4+} . The characteristic peak shifts 0.45 eV to lower energy, indicating lower valence state Mo in $1\text{T}''\text{-MoS}_2\text{-10.6\%}$.^{46, 47} The result is consistent with **Figure 2c**.

Comment 7. Mo L_3 edge can also help to distinguish the phase transition, please provide.

Response: Thanks for the Reviewer's suggestion. The XAFS data related to Mo L_3 edge are added to distinguish the different phase. As shown in **Figure R8**, the Mo L_3 -edge XANES spectra show slight difference between 2H-MoS_2 and $1\text{T}''\text{-MoS}_2\text{-10.6\%}$. For 2H-MoS_2 , the Mo L_3 -edge white line peak locates at 2524.01 eV and it is ascribed to the Mo^{4+} . Then, the characteristic peak shifts 0.45 eV to lower energy, indicating lower valence state Mo in $1\text{T}''\text{-MoS}_2\text{-10.6\%}$ (*Adv. Sci.* 2021, **8**, 2002768; *Nat. Comm.* 2021, **12**, 7195; *J. Am. Chem. Soc.* 2022, **144**, 4863-4873). The result is consistent with **Figure R7**. Moreover, the lower valence state means that the electron distribution around the Mo–Mo bonds in $1\text{T}''\text{-MoS}_2\text{-10.6\%}$ is more concentrated.

Figure R8. Mo L_3 -edges XANES spectra of $2\text{H-MoS}_2\text{-V}_\text{S}$ and $1\text{T}''\text{-MoS}_2\text{-10.6\%}$.

Change to the Manuscript (Page 9): Furthermore, the Mo L_3 -edge XANES spectra of $2\text{H-MoS}_2\text{-V}_\text{S}$ and $1\text{T}''\text{-MoS}_2\text{-10.6\%}$ were compared (**Supplementary Figure 9**). For $2\text{H-MoS}_2\text{-V}_\text{S}$, the Mo L_3 -edge white line peak locates at 2524.01 eV and it is ascribed

to the Mo^{4+} . The characteristic peak shifts 0.45 eV to lower energy, indicating lower valence state Mo in $1\text{T}''\text{-MoS}_2\text{-10.6\%}$.^{46, 47} The result is consistent with **Figure 2c**.

Comment 8. Figure 2f and Figure 2g the dpi is not clear enough for me to recognize the S vacancies. Could the author provide a clearer image? The axis for Figure 2g is important, please make it more clear for the Reviewers to recognize. Why does the author use ACTEM not HAADF-STEM to identify the S vacancy?

Response: The ACTEM image (**Figure R9**) in the original manuscript is indeed unclear enough to distinguish the S vacancies. We have provided a clearer enlarged image to recognize the S vacancies in the revised manuscript (*ACS. Ener. Lett.* 2019, **4**, 430–435). The axis for **Figure 2g** is labeled more clearly for the Reviewers to recognize. The HAADF-STEM image (**Figure R10**) shows that $1\text{T}''\text{-MoS}_2\text{-10.6\%}$ contains notable S vacancies (Red dotted circles). The total number of S-vacancies in **Figure R10** is about 15 (out of 54 Mo atoms, $\sim 10.6\%$ S-vacancies).

Figure R9. ACTEM observations of S vacancies in a $1\text{T}''\text{-MoS}_2\text{-10.6\%}$ sheet. The corresponding intensity distribution follows three different lines. Molybdenum atom has higher contrast than sulfur atom because of its high atomic number. The red dashed circles indicate a decrease in intensity due to the absence of S atoms.

Figure R10. HAADF-STEM image of 1T'-MoS₂ crystal with ~10.6% S-vacancy. The biggest and brightest dots are Mo atoms. Red dotted circles correspond to S-vacancies.

Comment 9. Figure 2e looks more like an FFT refined HRTEM / HAADF-STEM, please show the raw HAADF-STEM image here in case of distortion.

Response: Figure 2e is not an FFT refined HRTEM/HAADF-STEM, and the raw HAADF-STEM image in an enlarged area is provided (**Figure R11**). The SAED pattern is shown in **Supplementary Figure 5**.

Figure R11. (a) The raw HAADF-STEM image of 1T'-MoS₂-10.6% sheet.

Comment 10. With the increasing percentage of phase transition, the Tafel slope is not changing a lot while the onset potential is dramatically changing, can the author try to explain why?

Response: Thanks for the Reviewer's comments. The reasons are as follows: (i) the onset potential is more affected by the intrinsic activity of various 1T'-MoS₂-V_S

samples; (ii) 1T''-MoS₂-10.6% sample possesses the lowest overpotential of 158 mV at 10 mA cm⁻² and highest value of ECSA (25.1 mF cm⁻²), suggesting proper concentration of S vacancies can expose more electrochemically active sites and optimize the HER activity. According to the DFT calculations, when adsorbed atomic hydrogen is in a thermo-neutral state ($\Delta G_{H^*} \approx 0$), the proton-coupled electron transfer can be accelerated, and then molecular hydrogen will be released. Therefore, the onset potential achieves an optimal value. The Tafel equation is summarized from the experiment. The experimentally measured Tafel slope b is the apparent value of multi-step electrochemical reaction, and the slope corresponds to the apparent electron transfer number. Most electrochemical reactions involve multiple electron transfers, and the apparent electron transfer number includes the elementary rate determining step α^* and the electron transfer number n , so the Tafel slope of overall reaction should be $b = (2.303RT) / ((\alpha^* + n)F)$. In general, the difference of Tafel slope may be related to the electrolyte, crystal plane, test conditions, etc (*J. Phys. Chem.* 1993, **97**, 4769–4776; *J. Am. Chem. Soc.* 1999, **121**, 11855–11863; *Electro. acta.* 2007, **52**, 3493–3504; *J. Electro. Soc.* 2012, **159**, H864–H870; *ACS Catal.* 2014, **4**, 4364–4376).

Comment 11. The medium percentage of 1T'' phase of MoS₂ has the best EIS. The authors should comment on that.

Response: Thanks for your advice. The charge transfer resistance (R_{ct}) derived from EIS is used to evaluate the difficulty encountered when charge transfer across the electrode/solution interface, and it is greatly controlled by the reaction kinetics, the applied potential, the temperature, and the concentration of the reactant and reaction products. The HER process via proton-coupled electron transfer depends on both the electron transport in the catalyst and proton adsorption/desorption on the catalyst surface. S vacancies in 1T''-MoS₂ not only modify the electrical conductivity to facilitate electron transport but also tune the hydrogen adsorption free energy (ΔG_{H^*}) to optimize proton adsorption/desorption. Specifically, ΔG_{H^*} close to thermoneutral state supported by DFT calculation is realized in 1T''-MoS₂-10.6%, and the HER proceeds in an optimal condition to yield the smallest R_{ct} (*Nano Res.* 2021, **14**, 4814–

4821).

Comment 12. Figure 3g should use double Y to better illustrate the difference between Tafel slopes.

Response: We sincerely appreciate the Reviewer to give the valuable comments on our work.

Change to the Manuscript (Page 14): We have made revision accordingly in the revised manuscript (**Figure 3g**) as follows.

Figure R12. Overpotentials at current density of 10 mA cm⁻² of 1T^{''}-MoS₂-10.6% and 2H-MoS₂-V_S catalysts in comparison to that of other samples reported previously for HER catalysts in acidic electrolytes.

Comment 13. The author should also compare the K-weighted EXAFS with 2H MoS₂.

Response: Thanks for your comments. We have added the relevant data (**Figure R13**) in the revised Supporting Information.

Figure R13. k^3 -Weighted EXAFS oscillations of Mo foil, 1T''-MoS₂-2.0%, 1T''-MoS₂-10.6% and 2H-MoS₂.

Change to the Manuscript (Page 8–9): More S defects in 1T''-MoS₂-10.6% result in a decrease in Mo K-edge oscillation functions $k^3\chi(k)$ at the k range of 0–14 Å⁻¹ (**Supplementary Figure 8**) compared with 1T''-MoS₂-2.0% and 2H-MoS₂.

Reviewer: 2

Overall comment

Guo et al. present a study of vacancy-rich 1T'-MoS₂ as a HER catalyst, showing that the introduction of controlled vacancy concentrations can tune the electronic structure and hence the conductivity and HER activity. Overall, the conclusions appear to be well supported by the data, and the performance is impressive. The topic is also of broad interest to the community. I have a few comments:

Response: We sincerely appreciate the Reviewer to give the valuable comments on our manuscript, which are helpful to improve the quality of this manuscript. Elaborative revision was made carefully.

Comment 1. Although the experimental methods are described in the Methods section, the computational methods are not. The authors should clearly indicate factors such as DFT functional, energy cutoffs, etc., which can affect the predicted degree of charge localization and transfer. Likewise, the details of how ΔG^* was computed are missing.

Response: Thanks for your suggestions. The detail of the computational methods is added in the Supporting Information of the revised manuscript as follows.

Change to the Manuscript (Page 22–23): The Vienna Ab initio Simulation Package (VASP) was used to perform spin-polarized density functional theory (DFT) calculations, and the generalized gradient approximation of Perdew-Burke-Ernzerhof was introduced to describe electron exchange and correlation (*Comput. Mater. Sci.* 1996, **6**, 15–50; *Phys. Rev. B.* 1996, **54**, 11169–11186). The cut-off of plane-wave basis was set as 450 eV (*J. Am. Chem. Soc.* 2019, **141**, 790–793). The projector-augmented plane wave (PAW) was used to describe the electron-ion interactions (*Phys. Rev. B.* 1994, **50**, 17953–17979). The empirical dispersions of Grimme (DFT-D2) were applied to account for long-range van Waals interactions (*J. Comput. Chem.* 2006, **27**, 1787–1799). A set of (4×4×1) k-points were selected for geometric optimization, and the

convergence threshold was set as 10^{-4} eV in energy and 0.05 eV/Å in force, respectively. In the electronic structure calculation, denser k-points ($8 \times 8 \times 1$) were used for better accuracy. The vacuum slab of 15 Å was inserted in the z-direction for surface isolation to eliminate periodic interaction. The structure of $1T''$ -MoS₂ containing 36 atoms was introduced to model a system. The free energy of the adsorbed state was calculated as:

$$\Delta G = \Delta E_{H^*} + \Delta E_{ZPE} - T\Delta S$$

where ΔE_{H^*} is the hydrogen chemisorption energy, and ΔE_{ZPE} is the difference of the zero-point energy between the adsorbed state and the gas phase. Considering the fact that the vibration entropy of H^* in the adsorbed state is very small, the entropy of $1/2$ H₂ adsorption can be approximated as $\Delta S_H \approx -1/2 S_{H_2}^0$, where $S_{H_2}^0$ is the entropy of H₂ in the gas phase at the standard conditions.

Comment 2. The concept of self-regulation of charge, which the authors use in their title, is still a bit nebulous to me. Does this just mean tunability of electronic structure? In what sense is the material “self-regulating”? The term implies a capability to dynamically reorganize charge under different conditions (i.e., upon H adsorption), but I don’t see that reflected. This should be clarified.

Response: Thanks for the Reviewer’s valuable suggestions. The concept of self-regulation of charge reveals that the charge engineering mediated by S vacancy has an auto-optimizing effect on enhancing catalytic activity through regulating the distribution of active electronic states around the Fermi surface. In this work, it means tunability of electronic structure by introducing S-vacancies in $1T''$ -MoS₂. To clarify the phenomena, the correspondingly calculated charge density difference of different adsorbed S sites in $1T''$ -MoS₂ and $1T''$ -MoS₂-V1 is presented (**Figure R14**). It shows that intrinsic charge compensation from Mo–Mo bonds to neighboring S could manipulate the active electronic states of S atoms, allowing hydrogen to adsorb the active sites neither strongly nor weakly. The charge redistribution of S atoms around the V1 vacancy mainly promotes the formation of S...H bond with the optimal value $\Delta G_{H^*} = 0.098$ eV. The self-regulation via S vacancy mediated charge engineering is also

further demonstrated by the lowest overpotential of 1T^{'''}-MoS₂-10.6% (158 mV) compared with pristine 1T^{'''}-MoS₂ (283 mV).

Figure R14. (a, b) The correspondingly calculated charge density difference of different adsorbed S sites in 1T^{'''}-MoS₂ and 1T^{'''}-MoS₂-V1.

Comment 3. In many instances, 2D materials can exhibit vacancy clustering. In general, clustering is difficult to diagnose using the experimental techniques used by the authors, but it can be easily investigated theoretically. The models used by the authors assume isolated vacancies, but I would be interested to see whether the performance benefits are predicted to change at all upon formation of a divacancies or larger vacancy cluster. This could be related to the slight degradation in performance observed upon cycling, for instance.

Response: Thanks for the Reviewer's constructive comments. According to the Reviewer's suggestion, we calculate the corresponding ΔG_{H^*} of 1T^{'''}-MoS₂ with a divacancies (1T^{'''}-MoS₂-2V) to predict HER activity. As shown in **Figure R15**, it is found that the S atoms (S4*) connected with the activated Mo–Mo bonds exhibit stronger bonding strength, with the optimal value $\Delta G_{H^*}(S4^*) = 0.075$ eV. As expected, a divacancies with Mo–Mo bonds also synergistically regulates the ΔG_{H^*} of S atom compared to 1T^{'''}-MoS₂-V1 system. Thus, different configurations result in varied

activity, which may lead to the slight degradation in performance observed upon cycling.

Figure R15. (a, b) The ΔG_{H^*} in different exposed S atoms around Mo in 1T'-MoS₂-2V. The S_n* (n = 1-4) corresponds to the labeled number of S atoms.

Comment 4. The agreement between the theory and experiment for the optimal vacancy concentration is compelling. Although it's largely shown in the SI, I would recommend including in the main text a plot that directly compares the trend in predicted ΔG_{H^*} as a function of vacancy concentration with the trend in an experimental performance metric (e.g., Tafel slope, R_{ct} , or overpotential @ -10 mA/cm²) as a function of vacancy concentration. In principle, the calculations can be used to estimate the precise vacancy concentration that would lead to $\Delta G_{H^*}=0$, which could act as a guideline for optimizing future synthesis.

Response: Thanks for your suggestions. We have added a plot to compare the trend in predicted ΔG_{H^*} as a function of vacancy concentration with the trend in an experimental performance metric as follows. **Figure R16a** illustrates the relationship between ΔG_{H^*} and S vacancy concentrations (theoretically constructed), indicating charge compensation (from Mo–Mo bonds to neighboring S) effect on catalytic activity. The S vacancy modulated HER activity is further verified by the correlation between the overpotential, R_{ct} and S vacancy concentrations (experimentally measured). It is found that the lowest overpotential of 158 mV and minimum R_{ct} of 35.27 Ω (1T'-MoS₂-10.6%) is agreeable with the optimal ΔG_{H^*} predicted by theoretical calculations.

Figure R16. (a) The correlation between S vacancy concentrations and ΔG_{H^*} , (b) overpotential and R_{ct} in 1T^{'''}-MoS₂ with different S vacancy concentrations.

Change to the Manuscript (Page 17): Meanwhile, to illustrate the compelling agreement between the theory and experiment, the trend in predicted ΔG_{H^*} as a function of vacancy concentration is compared with the trend in experimental performance metrics (overpotential, R_{ct} and ECSAs). The optimal ΔG_{H^*} (0.098 eV) of the S-vacancy model (**Figure 4j**) is agreeable with the best performance of 1T^{'''}-MoS₂-10.6% (the lowest overpotential, minimum R_{ct} and highest ECSA in **Figure 4k, l**).

Comment 5. The calculations find one site in particular (S5* in Fig. 4) that becomes active upon vacancy incorporation. This suggests a 1:1 correspondence between vacancy concentration and active site density. It would be interesting to see a comparison of active site density (extracted from the capacitance measurement of the electrochemically active surface area) versus the vacancy density to see how well this relationship holds up.

Response: We appreciate the Reviewer's comments. The correlation between ESCA and vacancy density is plotted, and it first increases and then decreases (**Figure R17**). It is found that the largest ECSA (25.1 mF cm⁻²) is corresponding to the best HER activity when the vacancy density is 10.6%. It is revealed that the proper concentration of S vacancies significantly expands the ECSAs and exposes more electrochemically active sites. Therefore, the ECSAs of 1T^{'''}-

MoS₂-V1 system are qualitatively correlated with S vacancy density and HER activity to verify the charge regulation effect on catalytic activity.

Figure R17. ECSA and overpotential in 1T^{'''}-MoS₂ with different S vacancy concentrations.

Change to the Manuscript (Page 17): Meanwhile, to illustrate the compelling agreement between the theory and experiment, the trend in predicted ΔG_{H^*} as a function of vacancy concentration is compared with the trend in experimental performance metrics (overpotential, R_{ct} and ECSAs). The optimal ΔG_{H^*} (0.098 eV) of the S-vacancy model (**Figure 4j**) is agreeable with the best performance of 1T^{'''}-MoS₂-10.6% (the lowest overpotential, minimum R_{ct} and highest ECSA in **Figure 4k, l**).

Reviewer: 3

Overall comment

In this manuscript, the authors demonstrated a wet chemical etching method using $\text{K}_2\text{Cr}_2\text{O}_7$ to induce S vacancies in the 1T''- MoS_2 , which could result in the enhanced HER activity compared with the 2H- MoS_2 with S vacancy. It was found that the generated S vacancies would activate the Mo–Mo bonds in 1T''- MoS_2 to redistribute the electronic states of neighboring S atoms. The 1T''- MoS_2 -10.6% catalyst showed the best HER activity among all the tested samples, with an overpotential of 158 mV at the current density of 10 mA cm^{-2} and a Tafel slope of 74.5 mV dec^{-1} . Additionally, the mechanism of the enhanced HER activity was comprehensively studied and confirmed by the DFT calculations, supporting the authors' conclusions. Although this method is promising and may provide a new way for rational design and preparation of cost-effective and high-performance electrocatalyst, there are still several issues needed to be solved before this manuscript can be published.

Response: Thank you for your positive recognition. We are highly thankful for the Reviewer's appreciation on the novelty of a wet chemical etching method to induce S vacancies in the 1T''- MoS_2 . According to the suggestions and comments, careful revision has been made to the manuscript. All the corrections and amendments are highlighted in the revised manuscript.

Comment 1. It is noticed that the authors performed 60-min sonication of the catalyst solution before carrying out all the electrochemical tests. How can the metastable 1T''- MoS_2 -VS catalyst maintain the phase during such long-time sonication? Whether the 1T''- MoS_2 -VS is stable than the pure 1T''- MoS_2 ? The authors need to provide more experimental evidence to explain it.

Response: Thanks for your advice. The main factor affecting the stability of 1T''- MoS_2 is temperature. According to our previous work, the phase transition from 1T'' to 2H starts at about $130 \text{ }^\circ\text{C}$ (*J. Am. Chem. Soc.* 2019, **141**, 790–793). In this work, the

ultrasonic treatment was performed in ice water (0 °C) to maintain the phase stability of various 1T^{'''}-MoS₂-V_S samples. We selected 1T^{'''}-MoS₂-10.6% with the best performance and 1T^{'''}-MoS₂-22.9% with the highest S-vacancy concentration for XRD comparison before and after 60 min sonication. As shown in (**Figure R18**), the XRD patterns show that the 1T^{'''}-MoS₂-10.6% and 1T^{'''}-MoS₂-22.9% still maintain phase stability in the process of low-temperature ultrasound.

Figure R18. (a, b) XRD patterns of the 1T^{'''}-MoS₂-10.6% and 1T^{'''}-MoS₂-22.9% before and after 60 min sonication.

Comment 2. It is doubtful that the reference 2H-MoS₂-V_S sample was obtained by thermal treatment of 1T^{'''}-MoS₂-10.6%, how can the authors be sure that all the S vacancies would keep the same (amount and positions) as that in the original 1T^{'''}-MoS₂-10.6% sample?

Response: Thanks very much for your comments. 1T^{'''} and 2H-MoS₂ have different crystal structures. In the trigonal 1T^{'''} phase, the Mo–S coordination is octahedral, while in hexagonal 2H phase, the coordination is trigonal prismatic. The transformation of 1T^{'''} to 2H results in the change of Mo–S coordination geometry. Therefore, the S positions are different between the original 1T^{'''}-MoS₂-10.6% sample and the reference 2H-MoS₂-V_S sample. However, compare with 1T^{'''}-MoS₂-10.6%, the S vacancy concentrations did not change much in 2H-MoS₂-V_S. According to the XPS result (**Figure R19** and **Table R4**), the S vacancy concentration of 2H-MoS₂-V_S is about 11.3%. Therefore, there is almost no difference in S vacancy concentrations between

1T^{'''}-MoS₂-10.6% and 2H-MoS₂-V_S.

Figure R19. Mo 3d region and S 2s XPS spectra of the obtained samples: (a) 1T^{'''}-MoS₂-10.6%, (c) 2H-MoS₂-V_S. S 2p XPS spectra of the obtained samples: (b) 1T^{'''}-MoS₂-10.6%, (d) 2H-MoS₂-V_S.

Table R4. Mo/S ratio measured by XPS of 1T^{'''}-MoS₂-10.6% and 2H-MoS₂-V_S.

Sample	Mo Atom%	S Atom%
1T ^{'''} -MoS ₂ -10.6%	35.87	64.13
2H-MoS ₂ -V _S	36.05	63.95

Comment 3. The authors used the strong oxidizing agent to treat the samples to achieve the 1T^{'''}-MoS₂-V_S catalysts with S vacancies up to 22.9%. It is highly possible that oxidation would happen simultaneously. However, characterization of oxidation is absent in this paper. The authors need to use XPS, EDS, and other related characterizations to reveal it. If the samples were not oxidized, they need to explain the reason why only the S vacancies are introduced using this method.

Response: Thanks very much for your comments. We use strong oxidizing agent to obtain 1T^{'''}-MoS₂ and then introduce various concentrations of S vacancies. XPS

analysis presents the elemental composition (**Table R5**), indicating the existence of adsorbed oxygen species. **Figure R20** excludes the formation of hexavalent Mo, elemental S or high-valence S species. This conclusion is further supported by the SEM-EDS analysis (**Figure R21**). The reaction solutions ($\text{H}_2\text{SO}_4 + \text{K}_2\text{Cr}_2\text{O}_7$) after the preparation of 1T^{'''}-MoS₂-2.0% and 1T^{'''}-MoS₂-22.9% were subjected to the ICP-OES test. The measured content of Mo element is 0, suggesting that there is no loss of Mo during the etching process. Combining the above characterizations, it is demonstrated that only S vacancies are introduced and no oxidation happens simultaneously.

According to the previous studies, although the oxidation of MoS₂ basal plane is thermodynamically favorable, a relatively high kinetic barrier is faced (*J. Appl. Phys.* 2015, **115**, 135301; *2D Mater.* 2017, **4**, 025050; *Nat. Chem.* 2018, **10**, 1246–1251). The formation of oxidized MoO_x species usually needs high temperature, enough reaction time, or high concentration of oxidizing agents, *etc* (*Nat. Chem.* 2018, **10**, 1246–1251). 2H-MoS₂ electrocatalysts with high S vacancy concentrations (0% ~ 25%) prepared by mild argon plasma treatment, electrochemical desulfurization, *etc.* have been reported before (*Nat. Mater.* 2016, **15**, 48–53; *Nat. Comm.* 2017, **8**, 15113), and they were demonstrated to perform HER stably.

Table R5. Elemental composition of the representative catalysts from XPS.

catalyst	Mo (atomic. %)	S (atomic. %)	O (atomic. %)
1T ^{'''} -MoS ₂ -10.6%	34.57	61.78	3.65

Figure R20. XPS spectra (a) Mo 3d region and (b) S 2s region of 1T^{'''}-MoS₂-10.6%.

Figure R21. The SEM images and EDS analysis of 1T''-MoS₂-10.6%

Comment 4. In supplementary figure 4, 1T''-MoS₂-10.6% sheet shown in the TEM image seems no single crystal, but the SAED showed clear spot diffraction pattern from a single crystal, please check this data.

Response: Thanks for your comments. Because the as-prepared sample is a bulk material, we only select the edge of a large-size single crystal to harvest the spot diffraction pattern. A small-size thin sheet was subjected to TEM analysis, and it was demonstrated to be a single crystal by the bright SAED pattern (**Figure R22**).

Figure R22. HRTEM image of 1T''-MoS₂-10.6% thin sheet and the SAED pattern.

Comment 5. In supplementary figure 5, the authors showed the clear XRD peak shift between the 1T''-MoS₂-samples with different percentage of S vacancies. What is the origin for this peak shift? It is hard to believe that small percentage of S vacancies can

induce this peak shift. It may be caused by some oxidation or phase transition during the treating process. It would be better the authors can also show the Raman spectra of all the 1T^{''}-MoS₂-V_s samples.

Response: Thanks for your comments. As discussed above, oxidation or phase transition did not happen during the etching process. The enlarged XRD peaks of the various samples (**Supplementary Figure 6**) show that the (001) peak shifts to a lower angle with the etching time, indicating the increase of interlayer spacing. This may be caused by the weakening of interlayer van der Waals interaction after the generation of S-vacancies. The similar finding was also reported by others. Wang. *et al* presented the magnified XRD patterns of various 2H-MoS₂ samples, where the (002) peaks gradually shift to lower angles with the increase of S vacancy concentrations, indicating interlayer spacing expansion between S–Mo–S layers (*J. Am. Chem. Soc.* 2020, **142**, 4298–4308).

Comment 6. In supplementary figure 10, the DFT data show the results of samples with V_s=0%, 2.8%, 9.1%, 12.5%, 16.1%, which are different with the real data obtained in the prepared samples. Why not perform the calculations based on the same vacancy values to build models? If so, it will make the authors' conclusions more robust and convincing.

Response: We appreciate the Reviewer's valuable comments. For the as-prepared 1T^{''}-MoS₂-V_s samples, we used XPS spectra to determine Mo/S ratio and then the concentration of S vacancies experimentally. For the building theoretical models of S vacancies, it is slightly different for the real data obtained from the as-prepared samples. The vacancies tend to form cluster to stabilize the system, and clustered vacancies have lower formation energies than evenly dispersed ones (*Nat. Commun.* 2017, **8**, 15113). Once the first S-vacancy is formed, successive S-vacancies are more readily made in their vicinity. Thus, we theoretically constructed a series of S-vacancies such as single vacancy, three vacancies, four vacancies and five vacancies (corresponding to the S vacancy density of 2.8%, 9.1%, 12.5%, 16.1%). However, for the experimental samples, there are not only larger clustered vacancies but also some inevitably isolated S

vacancies. The slight difference does not affect the conclusion, and the theoretically predicted ΔG_{H^*} of 1T''-MoS₂-9.1% is nearly perfect to tally with the best chemical performance of 1T''-MoS₂-10.6%.

Comment 7. Another issue for the DFT calculation model is in Figure 4a, there are 3 S2, 2 S1 and 1 S3 sites in the draw scheme. Does that mean there are 3 types of S sites in the 1T''-MoS₂ structure, represented by S1, S2 and S3? Why the numbers are different?

Response: Yes, three types of S sites are in the 1T''-MoS₂ structure, according to the Crystallographic Data of 1T''-MoS₂ (**Table R6**). The crystal structure of 1T''-MoS₂ is determined by the single-crystal X-ray diffraction and it crystallizes in the trigonal space group *P*31m. The distortion of [MoS₆] octahedral coordination and the varied Mo–S bond lengths (2.370, 2.521, 2.459, and 2.298 Å) resulted in the different numbers of S sites (**Figure R23, Supplementary Table 1-2 and Table R7**).

Table R6. Crystallographic Data for 1T''-MoS₂.

Atom	x	y	z	Wyckoff.	Occ.
Mo	0.382 (6)	0.000	0.503 (15)	3c	1
S1	0.667	0.333	0.233 (3)	2b	1
S2	0.000	0.000	0.297 (4)	1a	1
S3	0.683 (18)	0.000	0.766 (11)	3c	1

Table R7. Selected bond distances (Å) for 1T''-MoS₂.

Bond	Bond distance (Å)	Bond	Bond distance (Å)
1. Mo—S1	2.371 (14)	4. Mo—S3 ⁱⁱ	2.521 (8)
2. Mo—S1 ⁱ	2.371 (14)	5. Mo—S3 ⁱⁱⁱ	2.521 (8)
3. Mo—S2	2.459 (12)	6. Mo—S3	2.298 (11)

Figure R23. The crystal structure of 1T''-MoS₂ with distorted octahedral coordination.

Comment 8. It seems that the concentration of S vacancy in the reported samples can be as high as 22.9%, can these samples maintain their structure during the test? What is the stability for all the 1T''-MoS₂-Vs samples with 2.0%, 7.7%, 10.6%, 17.9%, 22.9% S vacancies?

Response: Done. We supplemented the long-term durability tests of other samples and added relevant data in the revised manuscript. In **Figure R24**, CA analysis shows that the HER current of 1T''-MoS₂-2.0%/7.7%/17.9% is maintained over a 24 h period and 1T''-MoS₂-22.9% decreased slightly. No obvious change of all the 1T''-MoS₂-Vs samples is observed in Raman spectra before and after cycling test (**Figure R25**).

Figure R24. (a-d) The long-term durability tests for all the $1T''\text{-MoS}_2\text{-Vs}$ samples with 2.0%, 7.7%, 17.9%, 22.9% S vacancies.

Figure R25. (a-d) The Raman spectra of the $1T''\text{-MoS}_2\text{-Vs}$ samples with 2.0%, 7.7%, 17.9%, 22.9% S vacancies before and after cycling test.

Comment 9. The concentration of S vacancy is calculated based on the XPS results by converting the ratio of S and Mo. How did the authors exclude the existence of Mo vacancies? Moreover, there should be various types of S vacancies in the 1T''-MoS₂-Vs samples, how to identify them, and which kind of S vacancy has the best HER activity?

Response: Thanks for your comment. The STEM images of the four randomly chosen areas of the 1T''-MoS₂ crystal (**Figure R26**) show no Mo-vacancy (Mo removal). According to the ICP-OES analysis of the reaction solution after the etching process, no Mo loss is observed. Therefore, the existence of Mo vacancies can be excluded.

Different types of S sites in the 1T''-MoS₂ structure can induce various types of S vacancies (**Figure R27**) and even clustered vacancies. Since the as-prepared samples are bulk crystals, we only demonstrate the existence of S vacancies but challenging to identify different types. We attempted to obtain 1T''-MoS₂ monolayer by exfoliation of bulk crystal, which can be used to distinguish them. The relevant work about the preparation of high-quality few-layer or single-layer is ongoing.

The HER activity of S vacancies (V1, V2, V3 in **Figure 4d-f** and **Supplementary Figure 20**) themselves is unsatisfactory due to the unsuitable ΔG_{H^*} . Alternatively, after the generation of S vacancies, the charge redistribution of the activated Mo-Mo bonds and neighbouring S atoms (S1* ~ S5*) can regulate the hydrogen adsorption behaviour and then HER activity. Particularly, ΔG_{H^*} of the optimal active S5* atom is optimized to be 0.098 eV, which can facilitate HER efficiently.

Figure R26. (a-d) STEM image of the four randomly chosen areas of $1T''$ - MoS_2 crystal.

Figure R27. (a-c) The optimized structure of $1T''$ - MoS_2 -V1, $1T''$ - MoS_2 -V2 and T'' - MoS_2 -V3. (d-f) ΔG_{H^*} in different exposed S atoms around Mo in $1T''$ - MoS_2 -V1, $1T''$ - MoS_2 -V2 and T'' - MoS_2 -V3.

Comment 10. In Figure 2e and f, the HAADF-STEM image in Figure 2e is clearer than the Figure 2f, which is more suitable for the characterizations of S vacancies. The data obtained from Figure 2f are not convincing.

Response: Thanks for your suggestions. **Figure 2f** is indeed not as clear as **Figure 2e**, but it is paired with the line scan (**Figure 2g**) to identify S vacancies (*ACS Energy Lett.* 2019, **4**, 430–435). Lines 1 and 2 show two S gaps, while the area marked in line 3 shows no defects, indicating the presence of S vacancies in the 1T^{'''}-MoS₂-10.6%. The STEM images in **Figure R28** show that various 1T^{'''}-MoS₂ -Vs samples contain different concentrations of S-vacancies. We have added above data in the revised Supporting Information.

Figure R28. STEM image of 1T^{'''}-MoS₂ crystal with different S-vacancies (a) ~2.0% S-vacancy, (b) ~7.7% S-vacancy, (c) ~10.6% S-vacancy. The biggest and brightest dots are Mo atoms. Red dotted circles correspond to S-vacancies.

Comment 11. For the electrochemical test, the authors should also compare their results with pristine 1T^{'''}-MoS₂ and 2H-MoS₂ samples to make the conclusions more reliable and convincing.

Response: Done. We conducted supplementary experiments of commercial 2H-MoS₂ samples (**Figure R29**) for comparison. The commercial 2H-MoS₂ exhibits HER activity with overpotential of 410 mV at 10 mA cm⁻² and a Tafel slope of 175 mV dec⁻¹. And the value of ECSA is 1.17 mF cm⁻². In our preparation process, the pristine 1T^{'''}-MoS₂ sample contained a little amount of S vacancy (~ 2%) and it is selected as a reference sample.

Figure R29. HER performance in 0.5 M H₂SO₄. (a) Linear sweep polarization curves of commercial 2H-MoS₂. (b) The corresponding Tafel curves from the polarization curves. (c) Plots of current density difference against scan rates. (d) Nyquist plots of commercial 2H-MoS₂.

Change to the Manuscript (Page 11-13): To evaluate the influence of S vacancies on HER performance, the electrocatalytic activity of 1T^{'''}-MoS₂-V_S was studied in Ar-saturated 0.5 mol L⁻¹ H₂SO₄ solution. Linear sweep voltammetry (LSV) curves are shown in **Figure 3a** and **Supplementary Figure 14**. When the oxidation time increases up to 1 h, the overpotential continually decrease to 158 mV (1T^{'''}-MoS₂-10.6%), which agrees with ΔG_{H^*} of the theoretical calculations. To reveal the kinetic metrics, Tafel slope is used to investigate the rate-determining step for HER. As presented in **Figure 3b** and **Supplementary Figure 15**, the Tafel slope of 1T^{'''}-MoS₂-10.6% is 74.5 mV dec⁻¹, smaller than that of 2H-MoS₂-V_S. This suggests the pivotal role of activated S atoms for the absorption of hydrogen. Compared with 2H-MoS₂-V_S, 1T^{'''}-MoS₂-V_S have higher HER activity due to the self-regulation effect of Mo–Mo bonds. After activation by S vacancies, the Mo–Mo bonds enhance the S...H bonds by changing the electronic states of S atoms around Mo–Mo bonds, further leading the optimal ΔG_{H^*}

closer to 0. Accordingly, both experimental and theoretical results (**Supplementary Figure 14**) follow the similar trend that the 1T''-MoS₂-10.6% has the optimal performance. Hence, the combination of proper S vacancies and Mo–Mo bonds leads to higher HER activity of 1T''-MoS₂.

Electrochemical active surface areas (ECSAs) are estimated from the double-layer capacitance (C_{dl}) by measuring cyclic voltammetry (CV) curves. According to **Figure 3c** and **Supplementary Figure 15**, 1T''-MoS₂-10.6% possesses the highest value of ECSA (25.1 mF cm⁻²), considerably larger than those of 1T''-MoS₂-2.0% (15.2 mF cm⁻²), 1T''-MoS₂-7.7% (15.3 mF cm⁻²), 1T''-MoS₂-17.9% (23.6 mF cm⁻²), 1T''-MoS₂-22.9% (22.9 mF cm⁻²), commercial 2H-MoS₂ (1.17 mF cm⁻²) and 2H-MoS₂-V_S (0.97 mF cm⁻²), suggesting proper concentration of S vacancies can significantly expand the ECSAs and expose more electrochemically active sites. Besides that, the electrode kinetics was investigated at the hydrogen evolution voltage by electrochemical impedance spectroscopy (EIS). The result corresponds to charge transfer resistance of proton between electrode and electrolyte (**Figure 3d** and **Supplementary Figure 15**). 1T''-MoS₂-10.6% displays the lowest R_{ct} (35.27 Ω) among all the samples, indicating appropriate S vacancies can accelerate electrode kinetics for HER and reduce ohmic loss. S vacancies in 1T''-MoS₂ not only modify the electrical conductivity to facilitate electron transport but also tune the proton adsorption/desorption (in terms of ΔG_{H^*}) to optimize HER activity.

In summary, we highly appreciate the Reviewers' professional question, which helped us to make extensive investigations in detail to elaborate our points as well as to justify the significance of this work. We sincerely hope that the detailed explanations and analysis here could throw some light upon the novel aspects of this work. We really hope that our heartfelt explanation and extra experimental results in the updated manuscript could be rewarded an appreciation and reconsideration from the Reviewers.

REVIEWER COMMENTS

Reviewer #2 (Remarks to the Author):

The authors are to be commended on the careful and thorough responses to my questions and those of the other reviewers. I have no further concerns or reservations at this stage.

Reviewer #3 (Remarks to the Author):

The revised manuscript by Guo et al., has addressed some of the comments/questions raised by reviewers. However, there are still some points remaining unclear. I would like to see their further action before I can be fully convinced by their results.

1. I am not fully convinced by the authors that the "slight" difference of S vacancy density will not influence the result. As shown in Figure 3a, the 10.6% sample exhibited much better HER activity than the 7.7% sample with only 2.9% difference on the S vacancy density, which could not be well explained by the DFT calculation results.

2. Why is the A1g peak absent in the Raman spectra of 1T'-MoS₂ in Supplementary Figures 7 and 19? Such results are not consistent with the previous reports (*Nature Chemistry*, 2018, 10(6), 638–643; *Nature Materials*, 2018, 17(12), 1108-1114; *Advanced Materials*, 2019, 31(19), 1900568). If this is true, it may imply the existence of other phases in the as-prepared MoS₂ samples, rather than the 1T' phase. The authors should review these data and explain it.

3. In Supplementary Figure 12, it is obvious that the Mo 3d XPS peak shifted a lot when the S-vacancy increased from 2% to 22.9%. Same phenomenon is also presented in the S 2p XPS results. Such a large shift of the XPS peaks may indicate the phase change. The authors need to explain and discuss it in the manuscript.

4. The HAADF-STEM images are probably FFT filtered or processed with gaussian blur, which may not be appropriate for the following vacancy analysis. I would suggest the authors to use the raw HAADF-STEM images for the analysis.

5. For the electrochemical measurement, I would like to ask the authors to provide the calibration details of the reference electrode which is essential for HER data analysis. The r.p.m. value of the rotating disk working electrode used in the HER test should also be provided.

6. The HER activity is easily influenced by many factors such as the mass transport, ohmic losses, quality of the electrode and catalyst coatings. It is highly recommended that multiple electrodes are prepared for any RDE testing (e.g., minimum of 3 electrodes per experiment). The HER data with an appropriate error bar should also be presented.

Point-to-point Responses

We thank the editor for giving us a chance to further revise the manuscript, and also appreciate the reviewers' constructive comments which would improve our work in depth. After careful evaluation of the comments, complementary experiments, characterizations, calculations and discussions are made. We hope the response could meet the approval of both editor and reviewers.

The whole comment has been replied accordingly with changes clearly and highlighted. Please find point-to-point responses to the comments attached below.

Reviewer: 1

Overall comment

Reviewer 1 has no public concerns, they privately recommend publication with a minor request to on Figure R10 to move the raw figure to the right to better visualize the S vacancies.

Response: Thanks for the Reviewer's constructive comments. According to the Reviewer's suggestion, we have added the HAADF-STEM image of 1T'-MoS₂ crystal with ~10.6% S-vacancy to better visualize the S vacancies in the revised manuscript (Figure 2g).

Figure R1. Characterizations of structure of 1T'''-MoS₂-V_S. (a) Fabrication procedure of 1T'''-MoS₂-V_S crystals synthesized by high temperature solid-state reactions for chemical insertion and wet chemical routes for K⁺ extraction and S vacancies generation). (b) Raman spectra of 2H-MoS₂-V_S, 1T'''-MoS₂-2.0% and 1T'''-MoS₂-10.6%. (c) Mo K-edge XANES spectra and (d) FT-EXAFS spectra of Mo foil, 2H-MoS₂, 1T'''-MoS₂-2.0% and 1T'''-MoS₂-10.6%. (e) HAADF-STEM image of 1T'''-MoS₂-10.6% sheet. (f) ACTEM observations of S vacancies in a 1T'''-MoS₂-10.6% sheet. The corresponding intensity distribution follows three different lines. Molybdenum atom has higher contrast than sulfur atom because of its high atomic number. The red dashed circles indicate a decrease in intensity due to the absence of S atoms. (g) HAADF-STEM image of 1T'''-MoS₂ crystal with ~10.6% S-vacancy. The biggest and brightest dots are Mo atoms. Red dotted circles correspond to S-vacancies.

Change to the Manuscript (Page 10): The HAADF-STEM image of 1T'''-MoS₂-10.6% (Figure 2g) confirms the successful formation of S-vacancies with corresponding concentrations.

Reviewer: 2

Overall comment

The authors are to be commended on the careful and thorough responses to my questions and those of the other reviewers. I have no further concerns or reservations at this stage.

Response: Thanks very much for the Reviewer's approval.

Reviewer: 3

Overall comment

The revised manuscript by Guo et al., has addressed some of the comments/questions raised by reviewers. However, there are still some points remaining unclear. I would like to see their further action before I can be fully convinced by their results.

Response: We really appreciate the Reviewer to give the valuable comments, which are helpful to improve the quality of our manuscript. Elaborative revision was made carefully.

Comment 1. I am not fully convinced by the authors that the "slight" difference of S vacancy density will not influence the result. As shown in Figure 3a, the 10.6% sample exhibited much better HER activity than the 7.7% sample with only 2.9% difference on the S vacancy density, which could not be well explained by the DFT calculation results.

Response: We sincerely appreciate the Reviewer for giving us the constructive and professional comments to make our research work more convinced and precise. According to the Reviewer's concerns, we re-build the models and perform the HER calculations based on the experimentally measured vacancy concentrations. For the building theoretical models of S vacancies, the $1T''\text{-MoS}_2$ supercell ($2 \times 3 \text{ MoS}_2$ unit cell) containing 54 atoms is introduced to model a system where one, four, five, eight and ten atoms are removed (corresponding to the S vacancy concentration of 1.8%, 8%, 10.2%, 17.39%, 22.72%), nearly approaching the experimental values (2%, 7.7%, 10.6%, 17.9%, 22.9%). The vacancies tend to form cluster to stabilize the system, and clustered vacancies have lower formation energies than evenly dispersed ones. Once the first S-vacancy is formed, successive S-vacancies are more readily made in their vicinity (Nature Communications, 2017, 8, 15113). Compared to pristine $1T''\text{-MoS}_2$, due to the increase of S vacancy concentrations, the hydrogen evolution free energies

(ΔG_{H^*}) of 1T''-MoS₂-V_n system exhibit the stronger bonding strength ranging from 0.605 to -0.223 eV (Figure R2). The optimal value of ΔG_{H^*} (-0.024 eV) of 1T''-MoS₂-10.20 % is close to zero, where adsorbed atomic hydrogen is in a thermo-neutral state. The proton-coupled electron transfer can be accelerated, and then molecular hydrogen will be released. Thus, with the increase of S vacancies, the activated Mo–Mo bonds would further redistribute the active electronic states of 1T''-MoS₂-V_n system to improve hydrogen adsorption. Therefore, following the Reviewer’s suggestions, the complementary calculations authentically make our research work more convincing and robust. Figure R2b replaces the corresponding Supplementary Figure 14b.

Figure R2. Correlation of HER activity of 1T''-MoS₂-V_S with different S vacancies between experiment and calculation. (a) Linear sweep polarization curves of 1T''-MoS₂-V_S (V_S = 2.0%, 7.7%, 10.6%, 17.9%, 22.9%). (b) ΔG_{H^*} vs. the reaction coordination of HER for the S vacancies range of 0-22.72%.

Comment 2. Why is the A1g peak absent in the Raman spectra of 1T'-MoS₂ in Supplementary Figures 7 and 19? Such results are not consistent with the previous reports (Nature Chemistry, 2018, 10(6), 638–643; Nature Materials, 2018, 17(12), 1108-1114; Advanced Materials, 2019, 31(19), 1900568). If this is true, it may imply the existence of other phases in the as-prepared MoS₂ samples, rather than the 1T' phase. The authors should review these data and explain it.

Response: Thanks for the Reviewer’s constructive comments. The Raman spectrum of

1T'-MoS₂ has been carefully examined. Compared with the two phonon characteristics of A_{1g} mode, J₁, J₂ and J₃ modes in Raman spectrum of metastable phase 1T'-MoS₂ structure have different frequencies and greater intensity. The presence of these peaks is attributed to a zone-folding mechanism resulting from the formation of superlattice in the metastable MoS₂ (Physical Review B, 1991, 44, 3955). So, the peak vibration intensity of A_{1g} mode is relatively weak. By adjusting the intensity of the laser, the Raman spectra of 1T'-MoS₂ were tested repeatedly and the corresponding A_{1g} peak appears in **Figure R3a**. The main peaks at 15.08°, 31.34° and 34.35° in the XRD pattern are assigned to the (200), (-202) and (002) facets of the 1T'-MoS₂ (**Figure R3b**). The literatures (Nature Chemistry, 2018, 10(6), 638–643; Nature Materials, 2018, 17(12), 1108-1114; Advanced Materials, 2019, 31(19), 1900568) related to the Raman spectra of 1T'-MoS₂ have been cited in the revised manuscript. These pioneering works illustrated the different Raman vibration modes, which are helpful to our study on various phase structures of metastable MoS₂.

Figure R3. (a) Raman spectra of 1T'-MoS₂ and (b) XRD pattern of the 1T'-MoS₂.

Comment 3. *In Supplementary Figure 12, it is obvious that the Mo 3d XPS peak shifted a lot when the S-vacancy increased from 2% to 22.9%. Same phenomenon is also presented in the S2p XPS results. Such a large shift of the XPS peaks may indicate the phase change. The authors need to explain and discuss it in the manuscript.

Response: Thanks for the Reviewer's valuable suggestions. The change of S vacancy concentrations will lead to the change of valence state of Mo and S. Therefore, the binding energy of Mo 3d and S 2p in XPS (**Figure R4**) will have a shift of 0.1-0.8 eV, which is consistent with previous reports (Nature Materials. 2016, 15, 48–53; Nature Communications. 2017, 8, 15113; ACS Nano. 2019, 13, 6824–6834). Two peaks of 1T''-MoS₂-10.6% are located at 228.5 and 231.7 eV, corresponding to the Mo⁴⁺ 3d_{5/2} and 3d_{3/2}. The Mo 3d peaks of reference 2H-MoS₂-V_S are absent in 1T''-MoS₂-10.6%, confirming the purity of 1T'' phase. The S 2p spectrum of 1T''-MoS₂-10.6% shows two peaks at 163.3 and 162.15 eV, which belong to S 2p_{1/2} and S 2p_{3/2}, respectively. No peaks assigned to the oxidized Mo and S (hexavalent Mo, elemental S or high-valence S component) species appear. Compared to the reference 2H-MoS₂-V_S, S 2p orbital peaks of 1T''-MoS₂-10.6% shift 0.62 eV toward lower binding energies, suggesting the partial charges are transferred from the activated Mo–Mo bonds to neighbouring S atoms. In order to verify whether the phase structure has changed, XRD and Raman results are shown in the **Figure R5** below. The phase structures of 1T''-MoS₂-V_S (V_S=2.0%, 7.7%, 10.6%, 17.9% and 22.9%) did not change with the increase of S vacancy concentrations.

Figure R4. The XPS spectra of Mo 3d and S 2p of (a, f) 1T'''-MoS₂-2.0%, (b, g) 1T'''-MoS₂-7.7%, (c, h) 1T'''-MoS₂-10.6%, (d, i) 1T'''-MoS₂-17.9%, and (e, j) 1T'''-MoS₂-22.9%. (k) The S:Mo atomic ratio decreases with increasing duration of treatment time.

Figure R5. (a) XRD patterns and (b) Raman spectra of 1T''-MoS₂-V_S (V_S = 2.0%, 7.7%, 10.6%, 17.9% and 22.9%).

Comment 4. The HAADF-STEM images are probably FFT filtered or processed with gaussian blur, which may not be appropriate for the following vacancy analysis. I would suggest the authors to use the raw HAADF-STEM images for the analysis.

Response: Thanks for the Reviewer's suggestion. The HAADF-STEM images are not FFT filtered or processed with gaussian blur. Following your suggestion, we use the raw HAADF-STEM images to specifically analyze different S vacancy concentrations. The raw HAADF-STEM image in an enlarged area is provided in **Figure R6**.

Figure R6. The raw HAADF-STEM image of 1T''-MoS₂-10.6% sheet.

Comment 5. For the electrochemical measurement, I would like to ask the authors to provide the calibration details of the reference electrode which is essential for HER data analysis. The r.p.m. value of the rotating disk working electrode used in the HER test should also be provided.

Response: Thanks for the Reviewer's comments. More detailed calibration method of the reference electrode and electrochemical test details are necessary for analyzing HER data. It is more helpful to compare with other reported literatures. According to the Reviewer's suggestion, we have added the relevant data (see the Experimental Section in the Supporting Information for details) in the revised Supporting Information. The calibration of SCE reference electrode is performed in a standard three-electrode system with Pt wires as the working electrode, graphite rod as the counter electrode, and the SCE as the reference electrode. Electrolytes are pre-purged and saturated with high purity H_2 . CVs were run at a scan rate of 1 mV s^{-1} , and the average of the two potentials at which the current crossed zero was taken to be the thermodynamic potential for the HER (**Figure R7**). **Figure R7** was added to the revised supplementary information (**Supplementary Figure 24**). All the potentials were converted to the potential versus the reversible hydrogen electrode (RHE). In $0.5 \text{ M H}_2\text{SO}_4$, $E(\text{RHE}) = E(\text{SCE}) + 0.272 \text{ V}$. The rotating disk working electrode was rotated at 1600 rpm to remove the hydrogen gas bubbles formed at the catalyst surface.

Figure R7. The calibration curve of SCE reference electrode.

Change to the Manuscript (Page 23): The calibration of SCE reference electrode is performed in a standard three-electrode system with Pt wires as the working electrode, graphite rod as the counter electrode, and the SCE as the reference electrode. Electrolytes are pre-purged and saturated with high purity H₂. CVs were run at a scan rate of 1 mV s⁻¹, and the average of the two potentials at which the current crossed zero was taken to be the thermodynamic potential for the hydrogen electrode reactions. All the potentials were converted to the potential versus the reversible hydrogen electrode (RHE). In 0.5 M H₂SO₄, E(RHE) = E(SCE) + 0.272 V. The rotating disk working electrode was rotated at 1600 rpm to remove the hydrogen gas bubbles formed at the catalyst surface.

Comment 6. The HER activity is easily influenced by many factors such as the mass transport, ohmic losses, quality of the electrode and catalyst coatings. It is highly recommended that multiple electrodes are prepared for any RDE testing (e.g., minimum of 3 electrodes per experiment). The HER data with an appropriate error bar should also be presented.

Response: Thanks for your advice. According to the suggestions and comments, relevant reference experiments have been supplemented to the manuscript. In order to avoid the influence of the above factors on HER activity, the catalysts were dispersed in Nafion/alcohol solution (0.5 wt. %, Alfa Aesar) by sonication for 60 min in ice water. Then, 10 μL of the mixed solution was drop-casted onto a glassy carbon rotating disk working electrode (5 mm diameter) and dried with N₂. Moreover, three electrodes were prepared to test HER testing, which are labeled as electrode 1, electrode 2 and electrode 3, respectively. LSV curves are shown in **Figure R8**. The average values and standard deviation data of the overpotential of three electrodes tests have been added in **Figure R9** and **Table R1**. The average values of 1T"-MoS₂-2.0%, 1T"-MoS₂-7.7%, 1T"-MoS₂-10.6%, 1T"-MoS₂-17.9%, 1T"-MoS₂-22.9% and 2H-MoS₂-V_S are 285.33, 266.33, 158.33, 239.00, 246.00 and 364.67 mV, respectively. The standard deviation

values of 1T^{'''}-MoS₂-2.0%, 1T^{'''}-MoS₂-7.7%, 1T^{'''}-MoS₂-10.6%, 1T^{'''}-MoS₂-17.9%, 1T^{'''}-MoS₂-22.9% and 2H-MoS₂-V_S are 2.62, 3.09, 1.25, 2.16, 2.16 and 4.78 mV, respectively.

Figure R8. (a-f) LSV curves of 1T^{'''}-MoS₂-V_S (V_S = 2.0%, 7.7%, 10.6%, 17.9% and 22.9%) and 2H-MoS₂-V_S (three electrodes per experiment).

Figure R9. Average value of overpotentials and standard deviation of 1T'''-MoS₂-V_s (V_s = 2.0%, 7.7%, 10.6%, 17.9% and 22.9%) and 2H-MoS₂-V_s (three electrodes per experiment).

Table R1. Average value of overpotentials measured by three electrodes at current density of 10 mA cm⁻² of 1T'''-MoS₂-V_s (V_s = 2.0%, 7.7%, 10.6%, 17.9% and 22.9%) and 2H-MoS₂-V_s catalysts.

	Electrode 1 (mV)	Electrode 2 (mV)	Electrode 3 (mV)	Average values(mV)	Standard deviation(mV)
1T'''-MoS ₂ - 2.0%	289	284	283	285.33	2.62
1T'''-MoS ₂ - 7.7%	269	268	262	266.33	3.09
1T'''-MoS ₂ - 10.6%	160	157	158	158.33	1.25
1T'''-MoS ₂ - 17.9%	238	242	237	239.00	2.16
1T'''-MoS ₂ - 22.9%	248	247	243	246.00	2.16
2H-MoS ₂ - V _s	369	367	358	364.67	4.78

Change to the Manuscript (Page 23-24): In order to avoid the influence of the

above factors on HER activity, the catalysts were dispersed in Nafion/alcohol solution (0.5 wt. %, Alfa Aesar) by sonication for 60 min in ice water. Then, 10 μ L of the mixed solution was drop-casted onto a glassy carbon rotating disk working electrode (5 mm diameter) and dried with N_2 . Moreover, three electrodes were prepared to test HER testing, which are labeled as electrode 1, electrode 2 and electrode 3, respectively. LSV curves are shown in **Supplementary Figure 25**. The average values and standard deviation data of the overpotential of three electrodes tests have been added in **Supplementary Figure 26** and **Supplementary Table 6**. The average values of 1T^{'''}-MoS₂-2.0%, 1T^{'''}-MoS₂-7.7%, 1T^{'''}-MoS₂-10.6%, 1T^{'''}-MoS₂-17.9%, 1T^{'''}-MoS₂-22.9% and 2H-MoS₂-V_S are 285.33, 266.33, 158.33, 239.00, 246.00 and 364.67 mV, respectively. The standard deviation values of 1T^{'''}-MoS₂-2.0%, 1T^{'''}-MoS₂-7.7%, 1T^{'''}-MoS₂-10.6%, 1T^{'''}-MoS₂-17.9%, 1T^{'''}-MoS₂-22.9% and 2H-MoS₂-V_S are 2.62, 3.09, 1.25, 2.16, 2.16 and 4.78 mV, respectively.

REVIEWERS' COMMENTS

Reviewer #3 (Remarks to the Author):

The authors have provided new DFT calculation result and experimental data to support their conclusion. I recommend its publication.

Point-to-point Responses

Reviewer: 3

Overall comment

The authors have provided new DFT calculation result and experimental data to support their conclusion. I recommend its publication.

Response: Thanks very much for the Reviewer's approval.